# Transient capacitance changes recorded from vestibular type I hair cells are produced by $G_{K,L}$ gating and do not involve neurotransmitter exocytosis

Paolo Spaiardi[1,2,3] (ID), Roberta Giunta[1] (ID), Giorgio Rispoli[4] (ID), Sergio Masetto[1] and Stuart L. Johnson[5,6]

[1]*Department of Brain and Behavioural Sciences, University of Pavia, Pavia, Italy*
[2]*Department of Biology and Biotechnology, University of Pavia, Pavia, Italy*
[3]*Department of Physics, INFN – Pavia Section, Pavia, Italy*
[4]*Department of Neuroscience and Rehabilitation, University of Ferrara, Ferrara, Italy*
[5]*School of Biosciences, University of Sheffield, Sheffield, UK*
[6]*Neuroscience Institute, University of Sheffield, Sheffield, UK*

Handling Editors: Nathan Schoppa & Tina Pangršič

The peer review history is available in the Supporting information section of this article (https://doi.org/10.1113/JP288645#support-information-section).

**Abstract figure legend** Schematic diagrams of mammalian wild type (WT) (A) and *Caspr*-KO (B) type-I vestibular hair cells showing localization of the Caspr protein (orange), the $Ca^{2+}$ conductance (green) and the low-voltage-activated

S. Masetto and S. L. Johnson contributed equally to this work.

$K^+$ conductance (blue). In WT cells, Caspr is localized at the calyceal membrane, where it is essential for maintaining the junctional contact between the hair cell basolateral membrane and the afferent neuron. In *Caspr*-KO cells (B), the absence of Caspr results in a significant increase in the gap between the hair cell and the afferent nerve terminal, leading to a decreased resistance within the calyceal synaptic cleft. Simplified electrical circuits are overlaid on the hair cell diagrams in pink.

Whole-cell patch-clamp recordings of changes in membrane capacitance ($\Delta C_m$) reveal a large transient $\Delta C_m$ upon membrane hyperpolarization (C and D). The transient $\Delta C_m$, which is $Ca^{2+}$-independent and unrelated to vesicle fusion during exocytosis, is considerably smaller in *Caspr*-KO type-I hair cells compared to WT. This finding suggests that the high resistance of the calyceal cleft in WT cells may amplify the $\Delta C_m$ transient.

The amplitude of the transient $\Delta C_m$ closely correlates with the level of activation of the low-voltage activated outward rectifying $K^+$ current ($I_{K,L}$) (through the blue channel in A and B). This relationship indicates that the transient $\Delta C_m$ likely originates from charge mobilization during the gating of K,L channels (cross-sectional illustration in E), rather than from vesicle fusion events.

**Abstract** Head movements are detected and signalled to primary sensory neurons by vestibular types I and II hair cells. Signal transmission involves glutamate exocytosis from hair cells, which is triggered by $Ca^{2+}$ inflow through voltage-gated $Ca_V1.3$ $Ca^{2+}$ channels. In a previous study on mice, we reported a $Ca^{2+}$-dependent exocytosis in both hair cell types, measured as a sustained change in cell membrane capacitance ($\Delta C_m$) following cell depolarization, which was significantly smaller in type I than in type II hair cells. By contrast, only type I hair cells showed a large transient $\Delta C_m$, which was still present in $Ca_V1.3^{-/-}$ mouse type I hair cells. Here we investigated the nature of this transient $\Delta C_m$. We found that it was unaffected by 10 mM intracellular EGTA, which blocked most of the sustained exocytosis in these cells, demonstrating its insensitivity to intracellular $Ca^{2+}$. Moreover the amplitude of the transient $\Delta C_m$ correlated with the degree of activation of the low-voltage activated outward rectifying $K^+$ conductance, $G_{K,L}$, expressed by type I, but not type II hair cells. Finally the sign and kinetics of the transient $\Delta C_m$ changed based on voltage steps activating or deactivating $G_{K,L}$. These findings are consistent with the transient $\Delta C_m$ arising from the mobilization of charges during the gating of K,L channels, while excluding fast transient neurotransmitter exocytosis. Its large amplitude can be explained by the high resistance of the calyceal synaptic cleft since it was significantly reduced in $Caspr^{-/-}$ mice, which show a significantly larger synaptic cleft compared to wild type mice.

(Received 11 February 2025; accepted after revision 9 July 2025; first published online 26 August 2025)
**Corresponding authors** P. Spaiardi: Department of Brain and Behavioural Sciences, University of Pavia, Pavia, Italy. Email: paolo.spaiardi@unipv.it; S. L. Johnson: School of Biosciences, University of Sheffield, Sheffield, UK. Email: s.johnson@sheffield.ac.uk

## Key points

- Vestibular type I and type II hair cells signal head movement to the central nervous system.
- Signal transmission from both hair cell types relies on $Ca^{2+}$-dependent glutamate exocytosis, measured here as a sustained change in cell membrane capacitance ($\Delta C_m$). Type I hair cells exhibit also a large transient $\Delta C_m$, whose nature has not been elucidated.
- In this study we found that the transient $\Delta C_m$ does not involve exocytosis, but it is generated by the gating of the low-voltage activated outward rectifying $K^+$ conductance, specifically expressed in type I hair cells.
- Transient $\Delta C_m$ analysis (also carried out in mice lacking the core protein of the septate-like junction) conclusively demonstrates that type I hair cells, like type II ones, do not elicit a transient release of neurotransmitter.
- Knowledge of the basic mechanisms of vestibular signalling is crucial in the study of pharmacological treatment for vestibular disorders and in the drug side effects targeted there.

## Introduction

Vestibular organs of Amniotes are endowed with two types of sensory cells, called type I and type II hair cells. The most striking difference between type I and type II hair cells is their innervation. Although type II hair cells are contacted by several (10–20) small afferent and efferent nerve terminal boutons, type I hair cells are enclosed in a single giant afferent nerve terminal, called a calyx. The calyx synapse is a unique structure, characterized by both conventional quantal glutamatergic transmission (Bonsacquet et al., 2006; Matsubara et al., 1999; Sadeghi et al., 2014) and non-quantal afferent transmission ($K^+$ accumulation and ephaptic transmission; Contini et al., 2012, 2017, 2022, 2024; Govindaraju et al., 2023; Holt et al., 2007; Lim et al., 2011; Mukhopadhyay & Pangrsic, 2022; Songer & Eatock 2013; Yamashita & Omori, 1990). The membrane conductance of type I hair cells is dominated by $G_{K,L}$, a large $K^+$ conductance which involves the $K_V1.8$ subunit (Martin et al., 2024), and is functionally characterized by activating at unusually hyperpolarized membrane voltages ($V_m$, about −100 mV), being fully activated at around the cell resting $V_m$ (−60 mV: Rennie & Correia 1994; Rüsch & Eatock 1996). $G_{K,L}$ is not present in type II hair cells, and the afferent transmission at the type II hair cell bouton synapse has only been reported to be quantal, glutamatergic (Dulon et al., 2009; Spaiardi, Marcotti et al., 2020; Spaiardi, Tavazzani et al., 2020).

We recently investigated the exocytosis of synaptic vesicles in type I and type II mouse utricular hair cells by monitoring real-time changes in $\Delta C_m$ during whole-cell patch-clamp recordings (Spaiardi et al., 2022). With this technique electrical measurements of the $\Delta C_m$ of a single cell are used to follow the changes in cell-surface area associated with membrane addition during exocytosis, and membrane retrieval during endocytosis (Matthews & Fuchs 2010). We found that both type I and type II hair cells produce a sustained $\Delta C_m$ consistent with the exocytosis of synaptic vesicles triggered by the depolarization-induced inflow of $Ca^{2+}$ through the $Ca_V1.3$ $Ca^{2+}$ channels (Spaiardi et al., 2022). The sustained $\Delta C_m$ was approximately ten times smaller in type I hair cells compared to that in type II hair cells, suggesting that the exocytosis of synaptic vesicles was a much smaller component of signal transmission in the former (Spaiardi et al., 2022). However a large transient $\Delta C_m$ was always present upon repolarization to the holding $V_m$ of −81 mV from depolarizing voltage steps in type I, but not type II hair cells, the amplitude of which did not decrease following the largest depolarizing steps that elicit little or no $I_{Ca}$ (Spaiardi et al., 2022). Furthermore the transient $\Delta C_m$ was still present in $Ca_V1.3^{-/-}$ mice, which express a relatively small residual $I_{Ca}$ (19% compared to WT mice; Manca et al., 2021), whereas the sustained $\Delta C_m$ (exocytosis) was absent (Spaiardi et al., 2022). Finally the transient $\Delta C_m$ reversed direction upon repolarization to the holding $V_m$ of −81 mV from hyperpolarized voltages steps (Spaiardi et al., 2022). The nature of the transient $\Delta C_m$, however, was not elucidated, which is the aim of the present study.

We found that the transient $\Delta C_m$ in type I hair cells was not affected by intracellular $Ca^{2+}$ chelation by high EGTA, conclusively establishing its insensitivity to intracellular $Ca^{2+}$. Moreover, its amplitude, sign and kinetics correlated with $G_{K,L}$ activation or deactivation, suggesting it was generated by intramembrane charges translocation associated with gating of K,L channels. Finally we found that the amplitude of the transient $\Delta C_m$ was significantly reduced despite a similar $G_{K,L}$ in $Caspr^{-/-}$ mice, which because of the lack of the core protein of its septate-like junction, Caspr, at the type I hair cell-calyx synapse, have a much larger synaptic cleft (Sousa et al., 2009).

The present results are consistent with the afferent calyx enclosing a high-resistance intercellular compartment in series with the hair cell membrane resistance, which amplifies the capacitance signal generated by the gating of K,L channels. It cannot, however, exclude a physiological mechanism linking the K,L channel gating with post-synaptic elements by the Caspr protein, or by protein/s aggregated there, which could be not expressed, washed away or misplaced in the lack of this junction.

## Methods

### Ethical approval

Animal experimental work was licensed by the UK Home Office under the Animals (Scientific Procedures)

**Paolo Spaiardi** is a tenure track assistant professor in the Department of Biology and Biotechnology 'Lazzaro Spallanzani' at the University of Pavia. He completed his PhD in neuroscience and physiology at the same university. His research focuses on the neurophysiological and biophysical mechanisms underlying the functioning of the vestibular system and areas of the central nervous system, such as the parahippocampal cortices.

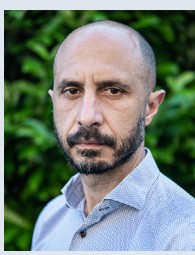

Act 1986 (PCC8E5E93 and PP1481074) and was approved by the University of Sheffield Ethical Review Committee (180626_Mar). Animals of either sex were killed by cervical dislocation followed by decapitation in accordance with UK Home Office regulations.

## Animals and tissue preparation

Vestibular hair cells were studied in acutely dissected C57B/6N mouse utricles from postnatal day 18 (P18) to P28, where the day of birth is P0. This is an age when the maturation of the sensory vestibular epithelium is considered complete (Burns & Stone, 2017; Burns et al., 2012). For some experiments utricles were obtained from *Caspr*$^{+/-}$ and *Caspr*$^{-/-}$ mice, aged P19-39. Caspr mice were a kind gift from Prof. Elior Peles, Weitzmann Institute of Science (Rehovot, Israel) (for details on the Caspr mice see Gollan et al., 2003).

Mouse utricles were dissected in the following extracellular solution (in mM): 135 NaCl, 5.8 KCl, 1.3 CaCl$_2$, 0.9 MgCl$_2$, 0.7 NaH$_2$PO4, 5.6 D-glucose, 10 Hepes-NaOH. Sodium pyruvate (2 mM), MEM amino acids solution (50×, without L-glutamine) and MEM vitamins solution (100×) were added from concentrates (Fisher Scientific, Loughborough Leicestershire, UK); the pH was adjusted to 7.5 (osmolality about 308 mmol/kg). The dissected utricles were transferred to a microscope chamber, immobilized using a nylon mesh fixed to a stainless-steel ring and continuously perfused with the above extracellular solution. The utricles were observed with an upright microscope (Nikon FN1, Tokyo, Japan) equipped with Nomarski differential interference contrast optics (X60 water immersion objective and X15 eyepieces).

## Whole-cell electrophysiology

All whole-cell patch-clamp recordings were performed at near body temperature (34°C–37°C) using an Optopatch amplifier (Cairn Research Ltd, Faversham, UK). Patch pipettes (3–4 MΩ) were pulled from soda glass capillaries (Hilgenberg, Malsfeld, Germany) and coated with surf wax (Mr. Zogs Sex Wax, Carpinteria, CA, USA) to minimize the fast capacitance transient of the patch pipette. Hair cells from the striola or extrastriolar regions of the mouse utricle were used in this study. Access to the hair cells was gained by using a 4 μm tip borosilicate glass pipette filled with a normal extracellular solution and connected to a syringe to apply light suction and pressure to clean the cell membrane prior to patching. As previously reported (e.g. Spaiardi et al., 2017) a patch pipette was used to remove the tissue debris above the targeted hair cell prior to seal it. The calyx had to be pierced to gain access to the hair cell basolateral membrane (see Fig. 1).

It is likely that both the outer and inner calyx membrane below the patch pipette were aspirated because of the negative pressure used to seal and to rupture the hair cell membrane. No substantial differences between WT and KO were noticed, possibly because the above procedure was rather variable among experiments.

For Ca$^{2+}$ current ($I_{Ca}$) and capacitance measurements (described below), K$^+$ currents were minimized using a CsGlutamate-based intracellular solution containing (in mM): 110 Cs-glutamate, 20 CsCl, 3 MgCl$_2$, 1 EGTA-CsOH, 5 Na$_2$ATP, 0.3 Na$_2$GTP, 5 Hepes-CsOH, 10 Na$_2$-phosphocreatine (pH 7.3 with CsOH; about 295 mmol/kg). Since $G_{K,L}$ is rather permeable to Cs$^+$ (Rennie & Correia 2000; Rüsch & Eatock, 1996; Spaiardi et al., 2017), this allowed us to identify type I hair cells since type II hair cells do not have $G_{K,L}$. The residual $I_{K,L}$ plus $I_h$, which are not blocked by intracellular Cs$^+$ (Fig. 7), were then blocked by locally perfusing the hair cells with an extracellular solution containing TEA and 4-AP (in mM): 110 NaCl, 5.8 CsCl, 1.3 CaCl$_2$, 0.9 MgCl$_2$, 0.7 NaH$_2$PO$_4$, 5.6 D-glucose, 10 Hepes, 30 mM TEA, and 15 mM 4-AP (pH adjusted to 7.5 with NaOH, osmolality about 312 mmol/kg).

Voltage protocols and data acquisition were controlled by pClamp software using a Digidata 1440A board (Molecular Devices, San Jose, CA, USA). Voltage-clamp recordings were low-pass filtered at 2.5 kHz (8-pole Bessel) and sampled at 5 kHz or 50 kHz. Data analysis was performed using Clampfit (Molecular Devices, USA) and Origin software (OriginLab, Northampton, MA, USA). Membrane potentials were corrected for the voltage drop across the series resistance ($R_s$) and a liquid junction potential of –11 mV between the Cs-Glutamate-based pipette solution and bath solution. The isolated Ca$^{2+}$ current recordings were corrected offline for the linear leak current ($I_{leak}$) typically calculated between –81 mV and –71 mV.

## Membrane capacitance measurements

Real-time measurement of cell membrane capacitance was performed with the 'track-in' circuitry of the Optopatch amplifier (Johnson et al., 2002, 2005) using a 4 kHz sine wave voltage command (13 mV RMS amplitude) applied at the holding $V_m$ of –81 mV, or –131 mV in some experiments. The exocytosis of synaptic vesicles was measured as the change $\Delta C_m$ produced by Ca$^{2+}$ influx elicited by 200 ms depolarizing voltage steps of variable size. The sine wave used to measure real-time $C_m$ was interrupted for the duration of the voltage steps. The capacitance signal from the Optopatch was amplified (50×), filtered at 250 Hz and sampled at 5 or 50 kHz. The $\Delta C_m$ as a function of cell membrane voltage was obtained as the difference between the mean baseline capacitance

signal and that measured over a 200 ms, or greater, period after each depolarizing voltage step. To investigate the $Ca^{2+}$-dependence of the $\Delta C_m$ changes (Fig. 2) we used the same CsGlutamate-based intracellular solution but increased the EGTA concentration to 10 mM, with an equimolar reduction in glutamate concentration.

### Statistical analysis

Differences in the mean were compared for statistical significance with an unpaired Student's two-tailed *t* test. For comparisons of multiple groups of data, we used a one-way ANOVA, or for two groups of multiple data sets we used a two-way ANOVA, both followed by a Sidak

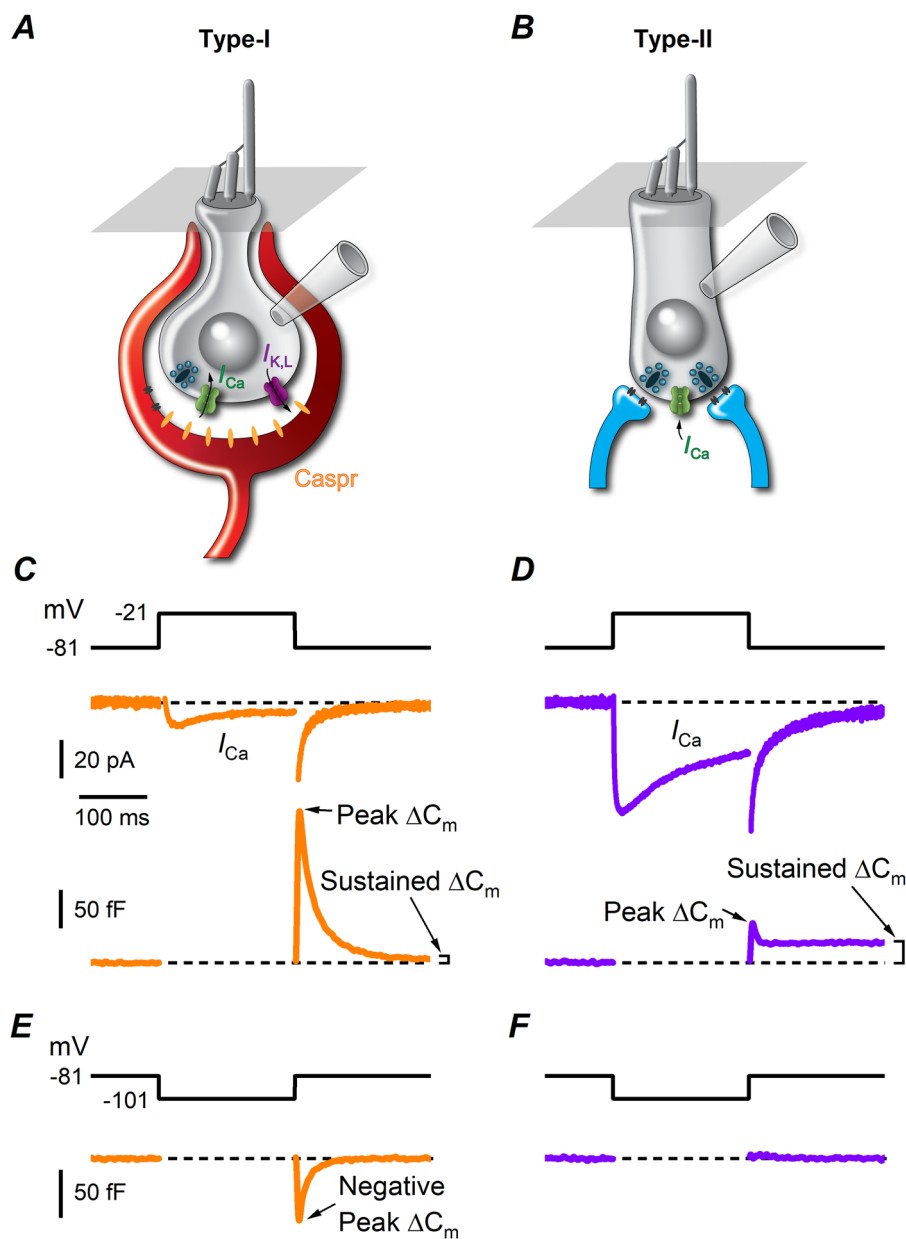

**Figure 1. The typical $I_{Ca}$ and $\Delta C_m$ recorded from a type I or a type II hair cell**
*A, B*, cartoon of the whole-cell recording configuration from a type I or a type II hair cell. Note that the afferent nerve calyx must be pierced to seal the hair cell basolateral membrane. *C, D*, representative voltage-clamp recordings with the CsGlutamate-based intracellular solution and the extracellular solution containing TEA and 4-AP to block $K^+$ channels (see Methods). From the $V_{hold}$ of −81 mV, the voltage steps to −21 mV elicited a small or large inward $I_{Ca}$ in type I or type II hair cells, while no current were elicited in response to the voltage step of −101 mV in either cell type. *E, F*, corresponding $\Delta C_m$ upon repolarization to holding $V_m$. While the sustained $\Delta C_m$ was significantly larger in type II than in type I hair cells, the latter showed a much larger transient $\Delta C_m$. Note that the sign of the transient $\Delta C_m$ elicited in type I hair cells reversed upon repolarization to holding $V_m$ from −101 mV.

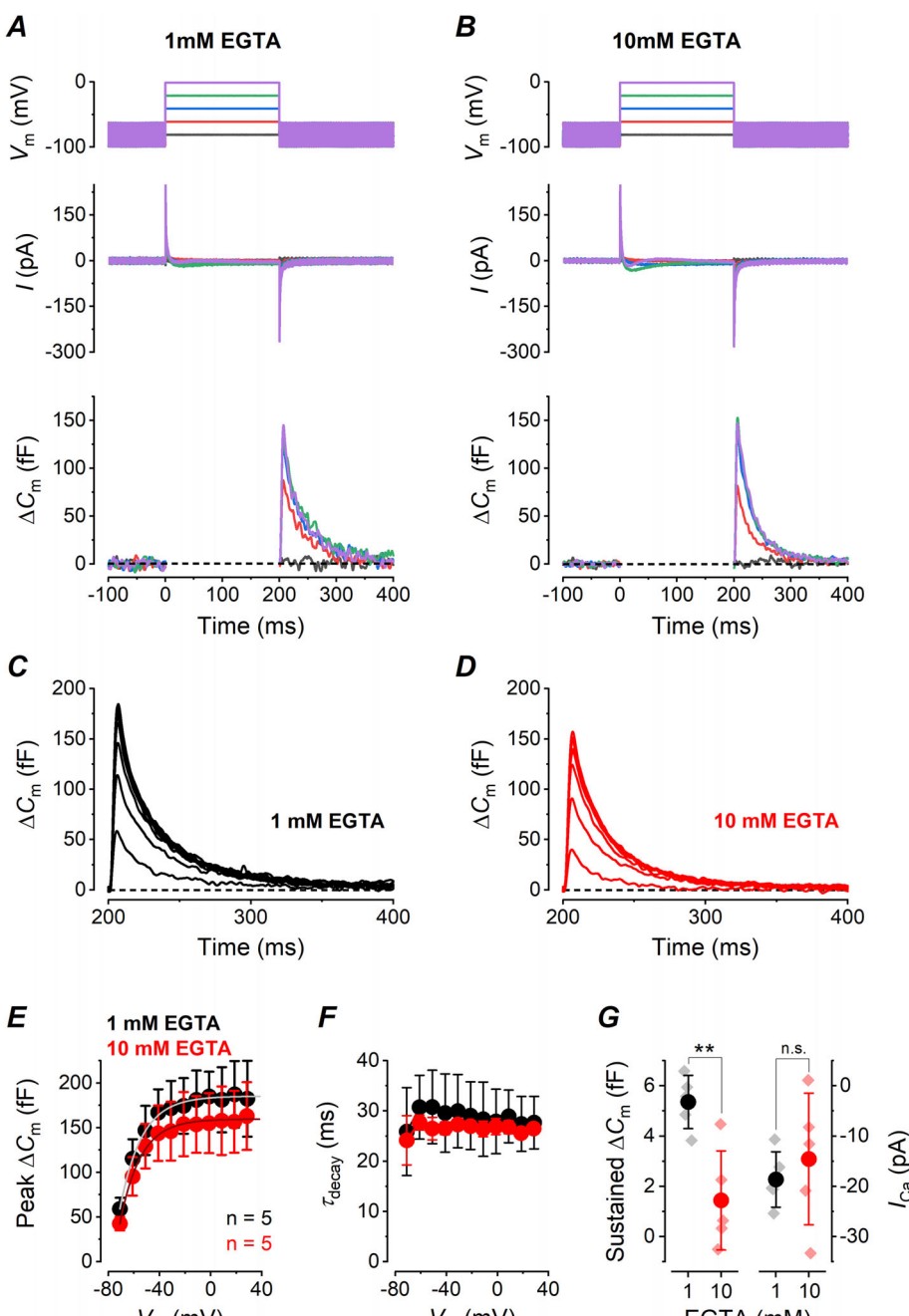

**Figure 2. Ca²⁺ current and exocytosis in wild-type (WT) vestibular type I hair cells of mature mouse utricles with 1 or 10 mM intracellular EGTA**

*A*, sample $I_{Ca}$ (top panel) and corresponding $\Delta C_m$ (bottom panel) responses recorded from a type I hair cell with 1 mM intracellular EGTA. Recordings were obtained in response to 10 mV voltage steps (200 ms) from the $V_{hold}$ of −81 mV ranging from −101 to 47 mV and returning to −81 mV, as from the voltage protocol shown above $I_{Ca}$ traces. The thick continuous line of the voltage protocol, interrupted for the duration of the voltage steps, consists of a 4 kHz sine wave *B*, sample $I_{Ca}$ (top panel) and corresponding $\Delta C_m$ (bottom panel) responses recorded from a type I hair cell with 10 mM intracellular EGTA. Same voltage protocol as in *A*. Despite the presence of $I_{Ca}$, the sustained $\Delta C_m$ responses were almost completely abolished by EGTA, but transients $\Delta C_m$ remained evident. *C*, average $\Delta C_m$ response recorded with 1 mM intracellular EGTA. *D*, average $\Delta C_m$ response recorded with 10 mM intracellular EGTA. $\Delta C_m$ recordings were performed in the presence of TEA and 4-AP. *E*, mean peak $\Delta C_m$ as a function of voltage in 1 or 10 mM EGTA. *F*, decay time constant of the peak transient $\Delta C_m$ in 1 mM or 10 mM EGTA. *G*, average sustained $\Delta C_m$ (re: left vertical axis) and $I_{Ca}$ (re: right vertical axis) measured in 1 or 10 mM EGTA in the recording pipette. Only values for steps from −81 mV to −21 mV are shown. The mean sustained $\Delta C_m$ resulted significantly smaller in 10 mM intracellular EGTA compared with 1 mM EGTA, whereas $I_{Ca}$ amplitude was similar in either condition.

multiple comparison post test. Mean values are quoted ± s.d.; $P < 0.05$ indicates statistical significance.

## Results

Vestibular hair cells are the sensory receptors of the vestibular system that convert head movements into neuronal activity via the release of neurotransmitter onto their synaptic contacts. The relation between the inward $Ca^{2+}$ current ($I_{Ca}$) and exocytosis has been investigated in mature vestibular type I and type II hair cells using single-cell patch-clamp recordings in *ex-vivo* explants of mouse vestibular organs (Dulon et al., 2009; Spaiardi, Marcotti et al., 2020; Spaiardi, Tavazzani et al., 2020; Spaiardi et al., 2022; Vincent et al., 2014). Although the exact functional role of type I and type II hair cells is currently uncertain, the cells are very different. The main features of type I and type II hair cells are summarized in Fig. 1. Type I hair cells are characterised by the presence of a single calyx afferent terminal that almost completely encases the basolateral membrane of the cell (Fig. 1A; Lysakowski & Goldberg, 2004; Wersäll, 1956). By contrast, type II hair cells are contacted by multiple small bouton-like afferent terminals that are like those on auditory hair cells (Fig. 1B; Lysakowski & Goldberg, 2004). Type I hair cells are also characterized by the expression of $I_{K,L}$ that is not present in type II hair cells (Correia & Lang, 1990; Rennie & Correia, 1994; Rüsch & Eatock, 1996). Both hair cell types express a voltage-dependent $I_{Ca}$, mainly carried by $Ca_V1.3$ $Ca^{2+}$ channels, which is much smaller in type I than in type II hair cells (Fig. 1C and D; Manca et al., 2021; Spaiardi et al., 2022). Consistent with their larger $I_{Ca}$, type II hair cells show a much larger amount of synaptic vesicle exocytosis than that seen in type I hair cells, which is evident from the 10 times larger sustained $\Delta C_m$ recorded in these cells (Fig. 1C and D). Type I hair cells, however, have a large positive transient $\Delta C_m$ component that is always present upon repolarization to the holding $V_m$ from depolarized potentials, which is much smaller in type II hair cells (Fig. 1C and D). The transient $\Delta C_m$ in type I hair cells becomes negative upon returning to the holding $V_m$ from voltage steps to more hyperpolarised potentials, which does not occur in type II hair cells (Fig. 1E and F). In contrast to the $Ca^{2+}$-dependence of the sustained $\Delta C_m$, which is indicative of synaptic vesicle exocytosis, the size of the transient $\Delta C_m$ did not vary based on $I_{Ca}$ amplitude and was still present, and apparently unaffected, in $Ca_V1.3^{-/-}$ mice (Spaiardi et al., 2022).

The above findings suggest that the large transient $\Delta C_m$ recorded from type I hair cells does not depend upon $Ca^{2+}$ entry through voltage-gated $Ca_V1.3$ $Ca^{2+}$ channels. However since a small residual $I_{Ca}$ was still present in $Ca_V1.3^{-/-}$ hair cells (Manca et al., 2021), whose molecular

nature remains to be elucidated, and to gain more information about the $Ca^{2+}$-dependence of $\Delta C_m$ in type I hair cells, we tested the effect of intracellular EGTA at a concentration of 1 and 10 mM. EGTA is a $Ca^{2+}$ chelator that has been widely used to probe the coupling between $Ca^{2+}$ channels and vesicular $Ca^{2+}$ sensors for neurotransmitter release (Augustine et al., 2003; Neher, 1998). Representative current traces and the corresponding $\Delta C_m$ obtained in the presence of either 1 or 10 mM intracellular EGTA from a type I hair cell are shown in Fig. 2A and B, respectively – the average $\Delta C_m$ from all cells are shown in Fig. 2C and D. The average values for the peak amplitude and decay time constant ($\tau_{decay}$) of the transient $\Delta C_m$ elicited upon repolarization to the holding $V_m$ of −81 mV following depolarized voltage steps in 1 and 10 mM EGTA are shown in Fig. 2E and F, respectively. Although there was an overall difference in the size of the peak transient $\Delta C_m$ and $\tau_{decay}$ at the two EGTA concentrations (two-way ANOVA; peak $\Delta C_m$ $P = 0.0002$; $\tau_{decay}$ $P = 0.0201$), post tests revealed no significant difference between values at individual $V_m$ (Sidak multiple comparisons post test $P > 0.05$ for each pair of values). Since we will show below that the transient $\Delta C_m$ correlates with $G_{K,L}$ activation, it is presumable that the above overall difference reflects the large variability in $G_{K,L}$ voltage-dependence among type I hair cells (Hurley et al., 2006). Maximal values for the peak $\Delta C_m$ were obtained by fitting the plot of $\Delta C_m$ *versus* $V_m$ with a single exponential function (Fig. 2E), and they were statistically similar in 1 mM and 10 mM EGTA (1 mM: 188 ± 33 fF, $n = 6$; 10 mM: 160 ± 35 fF, $n = 5$; $P = 0.21$ Student's unpaired *t* test). The independence of the transient $\Delta C_m$ from intracellular $Ca^{2+}$ indicates that it is related to something other than presynaptic vesicle fusion. By contrast, the sustained component of the $\Delta C_m$ response in type I hair cells, which is evident after the transient component, reflects synaptic vesicle exocytosis in these cells (Spaiardi et al., 2022). We therefore investigated the effect of high intracellular EGTA on exocytosis in these cells by measuring the sustained $\Delta C_m$ following a voltage step to −21 mV, which maximally activates $I_{Ca}$ (Spaiardi et al., 2022). While the size of the peak inward $I_{Ca}$ was not significantly different in 1 or 10 mM EGTA, the sustained $\Delta C_m$ was significantly reduced (Fig. 2G; $I_{Ca}$ $P = 0.54$; $\Delta C_m$ $P = 0.0045$, unpaired *t* test), indicating that synaptic vesicle exocytosis is largely uncoupled from $I_{Ca}$ in the presence of high intracellular EGTA. Since we measure exocytosis as the sustained $\Delta C_m$ towards the end of the recordings after the transient $\Delta C_m$, it is possible that this large transient component masks additional synaptic vesicle exocytosis within the first few 100 ms following the voltage steps, which we cannot discern. This aspect is important because a fast-transient exocytosis of glutamate might contribute to the phasic response recorded from calyx afferents innervating type I hair cells (Songer & Eatock, 2013). Although post-synaptic AMPA receptors,

which are expressed at the calyx terminal (Sadeghi et al., 2014), can generate transient postsynaptic responses because of their rapid desensitization, intrinsically transient glutamate exocytosis despite a sustained $I_{Ca}$ has been reported previously at the ribbon synapse of retinal bipolar cells (Singer & Diamond, 2003). To separate the synaptic from the non-synaptic component of the $\Delta C_m$ response, we subtracted the $\Delta C_m$ trace at a $V_m$ where $Ca^{2+}$-driven exocytosis was negligible from the one where it was maximal, but where the transient $\Delta C_m$ was approximately the same size and time course. In type I hair cells, the transient $\Delta C_m$ had approximately maximal amplitude and kinetics following voltage steps of $-41$ mV and above (Fig. 2E and F), while the peak $I_{Ca}$ and sustained $\Delta C_m$ occurred at $-21$ mV (Fig. 2G and Spaiardi et al., 2022).

Therefore we used the $\Delta C_m$ trace recorded after a voltage step to either $-41$ mV or $+19$ mV to subtract from that at $-21$ mV, since both potentials activate a much smaller $I_{Ca}$ and exocytosis in type I hair cells (Spaiardi et al., 2022) and are at opposite sides of the peak response at $-21$ mV, in 1 mM and 10 mM EGTA (Fig. 3A and B, respectively). The average $\Delta C_m$ resulting from the subtraction of the trace at $-41$ mV from that at $-21$ mV showed the absence of the large transient $\Delta C_m$ component in 1 mM (Fig. 3C) and 10 mM EGTA (Fig. 3D), while there was a sustained $\Delta C_m$ after the voltage step in 1 mM EGTA (Fig. 3C, $3.8 \pm 1.2$ fF, $n = 5$), that was not present in 10 mM EGTA (Fig. 3D, $0.2 \pm 1.9$ fF, $n = 5$: $P = 0.0066$, unpaired $t$ test). Similar results were obtained for the subtraction of the $\Delta C_m$ at $+19$ mV from $-21$ mV (Fig. 3E, 1 mM EGTA: $3.6 \pm 1.3$ fF, $n = 5$; Fig. 3F, 10 mM EGTA: $0.3 \pm 1.7$ fF, $n = 5$: $P = 0.0085$, unpaired $t$ test). The size of the sustained $\Delta C_m$ in 1 mM EGTA isolated from both subtractions was not significantly different from that measured after the transient $\Delta C_m$ (Fig. 2G) or from that previously reported (Spaiardi et al., 2022; $P = 0.6$, one-way ANOVA). Moreover the sustained $\Delta C_m$ in 10 mM EGTA obtained with and without subtraction

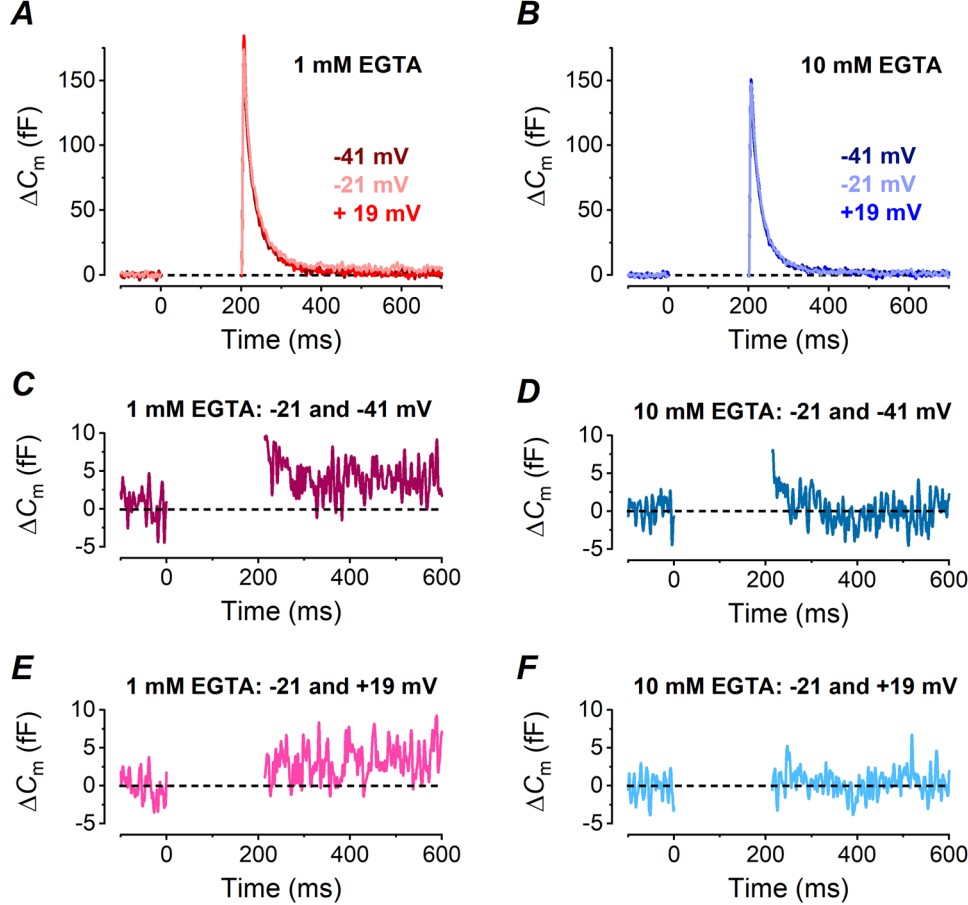

**Figure 3. Peak and sustained $\Delta C_m$ in wild-type (WT) vestibular type I hair cells are differently sensitive to 1 mM or 10 mM intracellular EGTA**
*A*, *B*, $\Delta C_m$ recorded after a voltage step to either $-41$, $-21$ or $+19$ mV in 1 or 10 mM intracellular EGTA, respectively. *C*, *D*, average differential trace obtained by subtracting $\Delta C_m$ recorded at $-41$ mV from that recorded at $-21$ mV in 1 mM or 10 mM intracellular EGTA, respectively. *E*, *F*, average differential trace obtained by subtracting $\Delta C_m$ recorded at $+19$ mV from that recorded at $-21$ mV in 1 mM or 10 mM intracellular EGTA, respectively.

were not significantly different as well ($P = 0.5$, one-way ANOVA). In conclusion the $\Delta C_m$ due to $Ca^{2+}$-dependent exocytosis in type I hair cells is fully uncoupled by 10 mM intracellular EGTA; moreover the large transient $\Delta C_m$ is non-synaptic, is not dependent on $Ca^{2+}$ entry and does not mask any exocytotic component that is larger than the one measured after the transient. Instead the size of the transient $\Delta C_m$ is non-linearly dependent on $V_m$ (Fig. 2*E*) and peaks in the negative direction in response to a return to the holding $V_m$ of $-81$ mV following a voltage step to a more hyperpolarized $V_m$ (Fig. 1*E*; Spaiardi et al., 2022). The negative transient $\Delta C_m$ is unlikely to correspond to endocytosis since this also requires elevations in $Ca^{2+}$ (Yamashita et al., 2010), which does not occur at these very negative potentials. Since the transient $\Delta C_m$ changes non-linearly with the $V_m$ and increased in size up to around $-41$ mV (Fig. 2*E*), a voltage at which $G_{K,L}$ activation nearly saturates (Spaiardi et al., 2020), it is possible that it is generated by the gating of $G_{K,L}$.

## The large transient $\Delta C_m$ in type I hair cells correlates with the level of $G_{K,L}$ activation

The main outward $K^+$ conductance in type I mammalian vestibular hair cells is $G_{K,L}$, which is absent from type II cells, activating at very negative $V_m$ ($-100$ mV) and being about half maximally activated at $-80$ mV (see Spaiardi et al., 2017 for characterization of $G_{K,L}$ activation curve). The $G_{K,L}$ channel gating transiently displaces charges from one side of the membrane to the other that sums with those displaced by the sinusoidal stimulus used to measure $C_m$ (see eqn (4) below) that could cause the transient increase in $C_m$ illustrated in Figs 1–3. This transient $\Delta C_m$ could be useful for investigating channel gating (or, in general, any protein rearrangement involving a charge displacement in a membrane protein) that can be especially prominent for ion channels characterized by prolonged tail currents after excitation with depolarizing pulses, as is the case for $G_{K,L}$. Several studies have taken advantage of $\Delta C_m$ measurements to detect changes in the number or mobility of any charged group residing within the electric field of the membrane, such as ion channel gating (Kilic & Lindau 2001) or membrane transporters (Lu et al., 1995). A large transient $\Delta C_m$, unrelated to $Ca^{2+}$-dependent exocytosis, has been reported in adrenal chromaffin cells and was shown to be due to $Na^+$ channel gating charge movement associated with channel de/inactivation (Horrigan & Bookman, 1994), the magnitude and time course of which were like that seen in type I vestibular hair cells. To investigate the relationship between the $G_{K,L}$ gating and the transient $\Delta C_m$, the latter was measured upon returning the same type I hair cell from different potentials to the holding $V_m$ of $-81$ mV, where $G_{K,L}$ is half activated (Fig. 4*A*, bottom panel), or

to $V_{hold}$ of $-131$ mV, where $G_{K,L}$ is fully deactivated (Fig. 4*B*, bottom panel). Average $\Delta C_m$ traces for the two holding voltages of $-81$ mV and $-131$ mV are shown in Fig. 4*C* and *D*, respectively. The curves describing the voltage dependence of the peak $\Delta C_m$ upon returning to $-81$ mV or $-131$ mV are shown in Fig. 4*E* (black or red dots, respectively). The peak $\Delta C_m$ responses elicited from $-131$ mV were similar to the $G_{K,L}$ activation curve (Fig. 4*E*, blue curve). This indicates that the transient $\Delta C_m$ could be generated by the translocation across the hair cell membrane of the voltage sensor gating charge, associated with K,L channel closure upon repolarizing to the $V_{hold}$ of $-131$ mV. Consistent with this hypothesis, the size of the $\Delta C_m$ transients from voltages more negative than $-70$ mV to $V_{hold}$ of $-131$ mV (Fig. 4*E*, red dots) was larger than the ones to $V_{hold}$ of $-81$ mV (Fig. 4*E*, black dots), due to the larger number of channels that close for the former voltage step (moving a larger number of gating charges) with respect to the latter. The values of the transient $\Delta C_m$ upon repolarization to either $V_{hold}$ of $-81$ mV or $-131$ mV from voltages are more depolarized than $-51$ mV overlap (Fig. 4*E*, black and red dots, respectively), because at these voltages $G_{K,L}$ is fully activated. As expected the amplitude of the transient $\Delta C_m$ saturates at voltages where all the gating charges of $G_{K,L}$ are supposedly in the open or in the closed positions. The presence of a transient $\Delta C_m$ at voltages where $G_{K,L}$ is not yet active ($-100$ mV) in Fig. 4*E* is not an incongruity, since $G_{K,L}$ could be well described by an allosteric Markov gating model showing multiple closed and open states (Spaiardi et al., 2017). According to this model the gating particles are expected to move at voltages just below the threshold for opening the channels, due to the redistribution of the channels among the closed states, resulting in a curve for the gating charge ($Q$), that is, in a curve for the amplitude of the transient $\Delta C_m$, that is, voltage, shifted to the left (i.e. toward more hyperpolarized voltages) compared to $G_{K,L}$ activation curve (Armstrong, 1981; see Catacuzzeno et al., 2023 for a recent review). On the contrary, the earlier saturation of the $Q$ curve in respect to the $G_{K,L}$ one for voltages between $-60$ and $-30$ mV is likely due to the relatively slow activation kinetics of $G_{K,L}$, since the 200 ms steps used here are too brief to let the $G_{K,L}$ to reach the steady-state activation (see, e.g. Fig. 7 in Spaiardi et al., 2017). Since all K,L channels are closed at $-131$ mV, there would be no gating charge movements for voltage steps to more negative potentials, and no transient $\Delta C_m$ was indeed recorded at $-141$ mV (Fig. 4*E*, red dots). Also since additional K,L channels will open upon stepping, for instance, from $-91$ mV to $-81$ mV, which will produce a movement of the gating charges in the opposite direction to their closure, a negative transient $\Delta C_m$ is expected, as it was indeed detected (Fig. 4*E*, black dots). Finally the $\tau_{decay}$ of the transient $\Delta C_m$ elicited upon repolarization from each test potential to the holding $V_m$ of $-131$ mV was

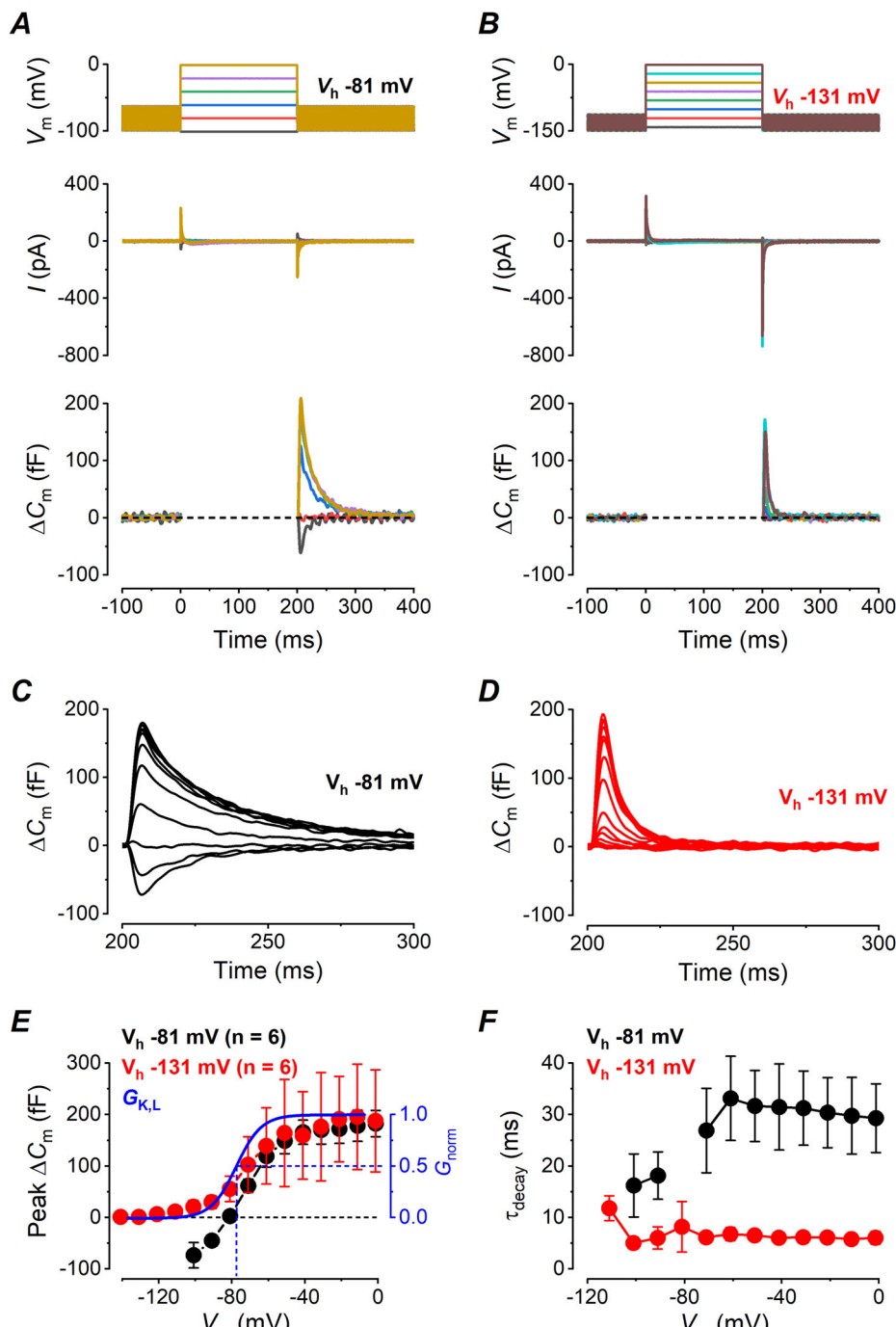

**Figure 4. Transient $\Delta C_m$ in wild-type (WT) vestibular type I hair cells of mature mouse utricles obtained by using different holding $V_m$**

*A*, sample $I_{Ca}$ (top panel) and corresponding $\Delta C_m$ (bottom panel) responses recorded from a type I hair cell in response to 20 mV voltage steps (200 ms) from holding $V_m$ of −81 mV ranging from −101 to −1 mV and returning to −81 mV, as from the voltage protocol shown above $I_{Ca}$ traces. *B*, sample $I_{Ca}$ (top panel) and corresponding $\Delta C_m$ (bottom panel) responses recorded from a type I hair cell in response to 20 mV voltage steps (200 ms) from holding $V_m$ of −131 mV ranging from −141 to −1 mV and returning to −131 mV, as from the voltage protocol shown above $I_{Ca}$ traces. *C*, average $\Delta C_m$ recorded upon repolarization to −81 mV following voltage steps ranging from −101 to −1 mV. *D*, average $\Delta C_m$ recorded upon repolarization to −131 mV following voltage steps ranging from −141 mV to −1 mV. *E*, voltage-dependence of the mean peak transient $\Delta C_m$ responses measured upon returning to the holding $V_m$ of −81 or −131 mV. The blue curve indicates the normalized activation curve for $G_{K,L}$, as from Spaiardi et al. (2017) (half activation voltage: −79.65 mV). *F*, mean $\tau_{decay}$ obtained by the same pool of cells as in *E*.

**Table 1. Parameters of equations**

| Abbreviation | Meaning | Value | Citations |
|---|---|---|---|
| $r_h$ | Cell radius | 2.5 μm | Govindaraju et al. (2022) |
| $s_c$ | Synaptic cleft length | 0.02 μm | Govindaraju et al. (2023) |
| $Vol_{cl}$ | Cleft volume | 1.2 μm³ | **eqn (6)** |
| $f$ | Fraction of basal membrane enwrapped by calyx | 0.75 | Govindaraju et al. (2023) |
| $\rho$ | Extracellular solution resistivity | 100 Ω cm | Textbook value |
| $R_m$ | Cell input resistance | ≈3 GΩ | Experimental data |
| $R_s$ | Series resistance | ≈10 MΩ | Experimental data |
| $R_c$ | Calyceal synaptic cleft resistance | ≈40 MΩ | Estimated |
| $C_m$ | Cell capacitance | ≈10 pF | Experimental data |
| $\Delta C_m$ | Cell capacitance change | | Variable |
| $I$ | Current | | Variable |
| $\Delta I$ | Current change | | Variable |
| $E_s$ | Voltage stimulus | | Variable |
| $\nu$ | Sinusoid frequency of $E_s$ | | Variable |
| $\theta$ | phase angle between $\Delta I$ and $V_s$ | | Variable |
| $n$ | change of cleft $K^+$ concentration moles/s | | Variable |
| $e$ | Elementary charge | $1.602 \times 10^{-19}$ C | Physical constant |
| $N_o$ | Avogadro number | $6.022 \times 10^{23}$ mol⁻¹ | Physical constant |

significantly faster than that elicited upon repolarization to −81 mV ($P < 0.0001$; two-way ANOVA; Fig. 4*F*), which is consistent with the faster decay of $I_{K,L}$ for stronger repolarizations (Spaiardi et al., 2017). All the results so far described strongly indicate that the transient $\Delta C_m$ can be considered, albeit within certain limits, a readout of the $G_{K,L}$ gating.

### Does the calyx play a role in the large transient $\Delta C_m$?

The large amplitude of the transient $\Delta C_m$ requires careful consideration of the relative amplitudes of the hair cell input resistance, $R_m$, and the series resistance, $R_s$, because large artefactual transient changes in membrane capacitance can be elicited by substantial changes in $R_m$ if $R_m$ and $R_s$ are of comparable amplitude (Barnett & Misler, 1997). Indeed with a standard $K^+$-based intracellular solution, $R_m$ in type I hair cells, because of the large $G_{K,L}$, is of similar amplitude to that of $R_s$ (around 10 MΩ; Contini et al., 2012). However substituting intracellular $K^+$ with $Cs^+$ and adding blockers of $K^+$ channels, as done here, increased the value of $R_m$ of type I hair cells to 3.06 ± 1.00 GΩ ($n = 25$, measured by a brief voltage pulse between −91 mV and −81 mV), similar to that of type II hair cells (2.56 ± 1.00 GΩ, $n = 25$, $P = 0.37$; Spaiardi et al., 2020b). Indeed even the most depolarized voltage steps elicited outward ($Cs^+$) current ≤ 100 pA in either cell type (Fig. 2*A* and *B*), meaning that $R_m$ was at all voltages ≥ 1 GΩ, i.e. much larger than the typical $R_s$ values. Therefore the dramatic difference in the amplitude of the transient $\Delta C_m$ (196.0 ± 46.8 fF *vs.* 32.7 ± 23.4 fF in the same type I and type II hair cells, measured at

−41 mV) cannot be explained by differences in $R_m$ or $R_s$ between the two hair cell types. However an additional $R_s$ should be considered here for recordings from type I hair cells, that is the resistance of the calyceal synaptic cleft, $R_c$, as follows.

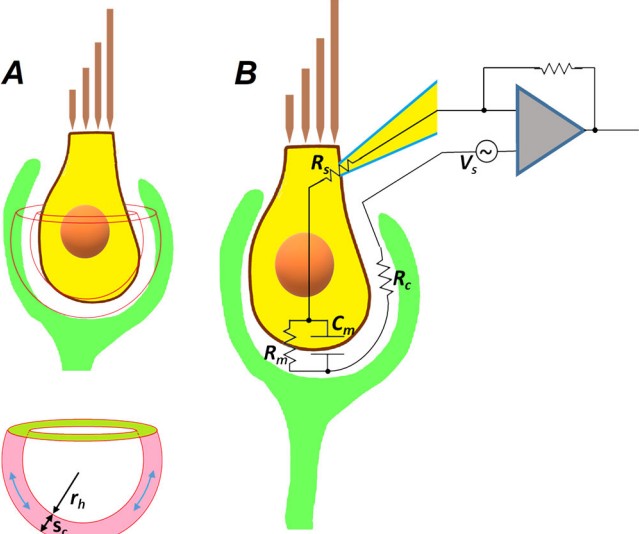

**Figure 5. The calyx cleft contribution to the hair cell electrical measurements**

*A*, geometry of the calyx: the hair cell basal pole is approximated to a sphere of radius $r_h$; $s_c$ is the thickness of the synaptic cleft (approximated to a sphere as well). *B*, the equivalent circuit model for a whole-cell recorded type I hair cell that measures current flowing in response to a voltage stimulus $V_s$, consisting of the series (access) resistance $R_s$, the cell membrane resistance $R_m$, the cell membrane capacitance $C_m$ and the cleft resistance $R_c$.

The basolateral membrane of type I hair cells is almost completely enveloped by the single giant afferent calyx nerve terminal (Figs 1*A* and 5), and early models (Goldberg, 1996) recognized that the elongated cleft space could limit the diffusion of ions and provide an increased electrical resistance.

The cleft region can be roughly represented by two hemispheric and concentric shells (Fig. 5*A*, top panel) of radiuses $r_h$ and $r_h+s_c$ (Fig. 5*A*, bottom panel), where $r_h$ is the hair cell radius (2.5 µm; Table 1) and $s_c$ is the synaptic cleft length (0.02 µm; data from Govindaraju et al., 2023), filled with the extracellular solution of resistivity $\rho$ (100 Ω·cm). The current flowing through the hair cell membrane is therefore thought to travel within the calyx cleft (blue arrows in the pink region), along an annular region (whose section is in green in Fig. 5*A* bottom panel). This corresponds to a resistance $R_c$ given by:

$$R_c = \rho \, \frac{2\pi r_h f}{\pi (r_h + s_c)^2 - \pi s_c^2} = \frac{\rho f}{s_c \left(1 + \frac{s_c}{r_h}\right)} \approx \frac{\rho f}{s_c} \quad (1)$$

where $f$ is the fraction of basolateral hair cell membrane (below tight junctions) enwrapped by calyx, that is between 0 (no calyx) and 1 (calyx wraps completely the hair cell): assuming $F = 0.75$ in eqn (1), it results in $R_c \approx 38$ MΩ. The resistance $R_c$ sums with $R_s$ in the capacitance measurements; therefore, if $Z$ is the impedance of the entire circuit, the current $I$ due to a sinusoidal voltage stimulus $V_s$ of angular frequency $\omega = 2\pi \nu$ (where $\nu$ is the sinusoid frequency) will be given by Ohm's law:

$$I = \frac{V_s}{Z} = \left(R_s + \frac{1}{\frac{1}{R_m} + j\omega C_m} + R_c\right)$$

$$V_s = \frac{\frac{1}{R_m} + j\omega C_m}{\frac{R_s + R_c}{R_m} + j\omega C_m (R_s + R_c) + 1} V_s \quad (2)$$

It is impossible to isolate a vector component of $I$ in eqn (2) that is directly proportional to the capacitance. However, since any change in $R_c$, $R_s$, $R_m$ or $C_m$ will produce a change $\Delta I$ of $I$, a $C_m$ change ($\Delta C_m$) can be estimated from the induced $\Delta I$ (eqn (2)), that was phase-shifted by $-90°$ with respect to the phase angle $\theta$ between $\Delta I$ and $V_s$. This $\Delta I$ is given by eqn (3):

$$\Delta I = \frac{\partial I}{\partial C_m} \Delta C_m$$

$$= \frac{j\omega}{\left[\frac{R_s + R_c}{R_m} + j\omega C_m (R_s + R_c) + 1\right]^2} V_s \Delta C_m \quad (3)$$

Since $R_m$ is at least an order of magnitude larger than $R_s + R_c$ (see above), the term:

$$\frac{R_s + R_c}{R_m}$$

can be neglected in eqn (3) that becomes:

$$\Delta I = \frac{j\omega}{\left[j\omega C_m (R_s + R_c) + 1\right]^2} V_s \Delta C_m \quad (4)$$

The phase angle $\theta$ of $\Delta I$ is given by:

$$\theta = \pi - 2\arctan\left[\omega C_m (R_s + R_c)\right] \quad (5)$$

Computer simulations of capacitance changes in whole-cell mode (in a simulated cell with $R_s = 10$ MΩ, $R_m = 1$ GΩ and $C_m = 6.5$ pF) showed that a 10 MΩ increase of $R_s$ gives an artefactual increase of $C_m$ of 50 fF (Fig. 2 of Santos-Sacchi, 2004). Here $R_c$ results in series with $R_s$ (Fig. 5*B*), i.e. $R_c$ just adds to $R_s$ (eqn (4) and 5); therefore, assuming an $R_c$ value about four times larger than $R_s$ (see above) would artefactually alter the $C_m$ measure significantly. Given the large value of $R_c$, an obvious question is what happens to the $C_m$ measures in the presence of an enlarged cleft calyx synapse: if $s_c$ increased, for example, by twofold (i.e. up to 0.04 µm), then $R_c$ would halve (from $\approx 38$ to $\approx 19$ MΩ; eqn (1)). Note that this artefact occurs also in recordings obtained with the Optopatch amplifier used here that can automatically correct the capacitance recordings of small changes in $R_s$ during the recordings, but it gives artefactually larger capacitance measurements when $R_s$ values approach the $R_m$ ones. In the calyx synapse, the apposed pre- and postsynaptic membranes are kept unusually close by a patterned alignment of proteins resembling a type of intercellular junction that is rare in vertebrates, the septate junction (Sousa et al., 2009). A core molecular component of the septate junction is Caspr, and in *Caspr*$^{-/-}$ mice the separation between the pre- and postsynaptic membranes at the calyx synapse is conspicuously irregular and often increased by an order of magnitude (Sousa et al., 2009). Therefore we measured the amplitude of the capacitive transient in *Caspr*$^{-/-}$ mice to check if it was affected by the $R_c$ reduction occurring in these conditions.

### The transient $\triangle C_m$ is smaller in *Caspr*$^{-/-}$ type I hair cells

We found that the amplitude of the transient $\Delta C_m$ in *Caspr*$^{-/-}$ type I hair cells was significantly smaller than in control (WT and heterozygous mice) cells at every $V_m$ tested ($P < 0.0001$ for pairs of values from $-51$ mV, Sidak multiple comparisons; Fig. 6*A*–*D*). In particular the maximal value for the peak $\Delta C_m$, obtained by exponential fitting of the $\Delta C_m$ *vs.* voltage plots, resulted larger in control type I hair cells (Fig. 6*E*, black dots; $193.2 \pm 39.8$ pF, $n = 6$) than in *Caspr*$^{-/-}$ cells (Fig. 6*E*, red dots; $121.9 \pm 23.6$ pF, $n = 11$; $P = 0.0003$, unpaired $t$ test). The $\tau_{decay}$ values of the transient $\Delta C_m$ for both control and *Caspr*$^{-/-}$ were negligibly voltage-dependent and over-lapped at every $V_m$ tested (Fig. 6*F*; $P = 0.8$, two-way

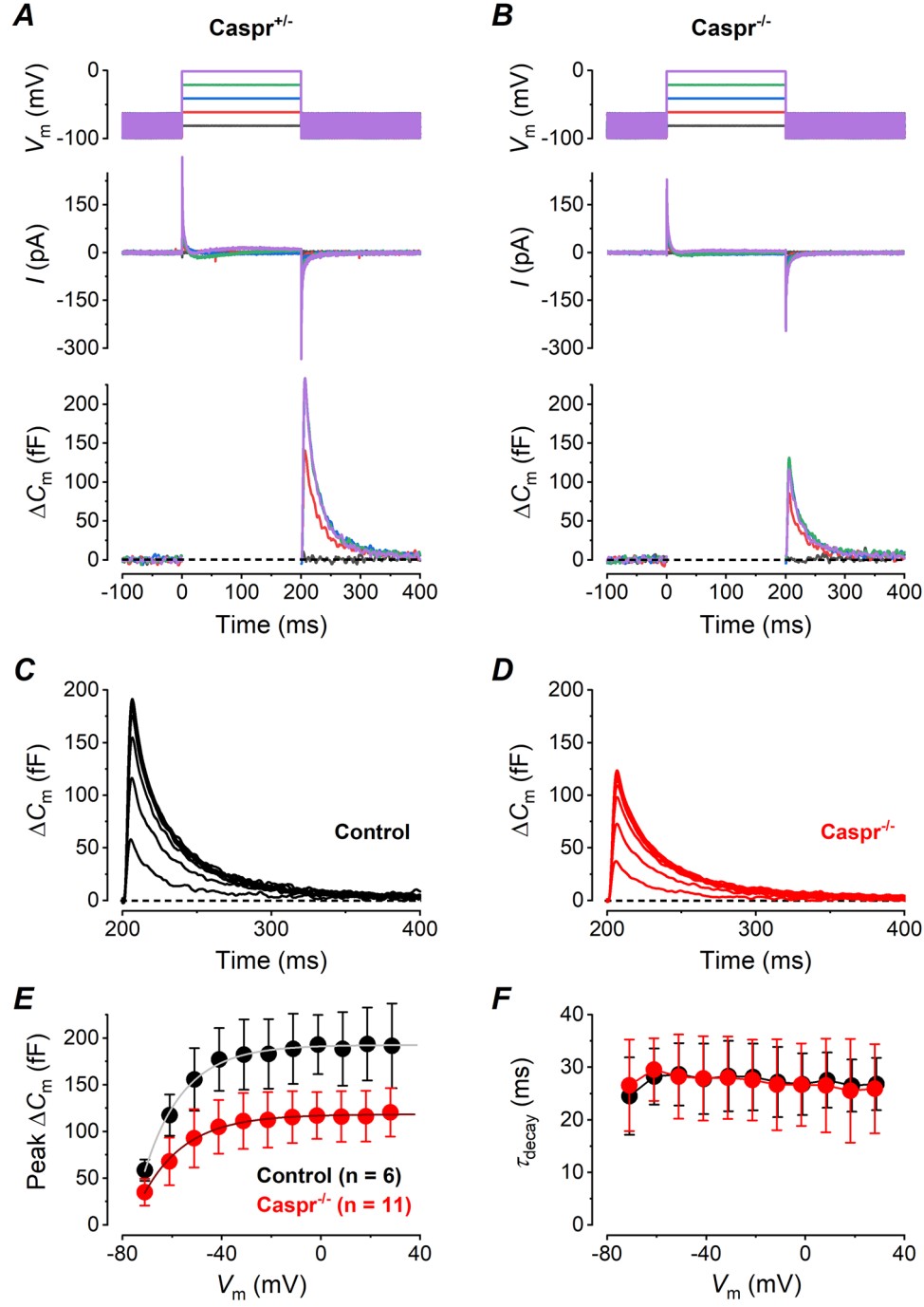

**Figure 6. Comparison between control and *Caspr*$^{-/-}$ type I hair cells of mature mouse utricles**

*A*, *B*, representative $I_{Ca}$ (top panel) and $\Delta C_m$ (lower panel) recordings from a type I hair cell of a control (*Caspr*$^{+/-}$) and a KO (*Caspr*$^{-/-}$) mouse, respectively. Voltage steps were from −81 to −1 mV in 20 mV increments, from a holding voltage of −81 mV. *C*, *D*, mean transient $\Delta C_m$ obtained from six control and 11 *Caspr*$^{-/-}$ type I hair cells, respectively. *E*, plot of the mean transient $\Delta C_m$ amplitude *versus* voltage (six control and 11 *Caspr*$^{-/-}$ cells). *F*, mean $\tau_{decay}$ of the transient $\Delta C_m$ *versus* $V_m$ (six control and 11 *Caspr*$^{-/-}$ cells). Caspr$^{-/-}$ cells (*n* = 11) were from 4 KO-mice. Control cells were from 1 Caspr Het mouse (*n* = 2) and two wild type mice (*n* = 4).

ANOVA, $P > 0.05$ for each pair of values, Sidak multiple comparisons).

The smaller peak transient $\Delta C_m$ responses of $Caspr^{-/-}$ type I hair cells could be due to a smaller number of K,L channels in these cells compared to controls. To investigate this possibility, we measured the amplitude of the K,L current in the presence of Cs-Glutamate in the intracellular solution before we perfused additional $K^+$ channel blockers to largely reduce this component. In both $Caspr^{-/-}$ and control type I hair cells, upon stepping to about $-120$ mV there was an initial inward peak current flowing through open K,L channels which then closed (Fig. 7$A$,$B$). From this $V_m$, the current was inward up to $-50$ mV and became outward for potentials positive to $-40$ mV, due to the mixed $Cs^+/K^+$ current reversal equilibrium in these experimental conditions (Spaiardi et al., 2017). The size of the current through $G_{K,L}$ was similar between $Caspr^{-/-}$ and control cells for voltages between $-91$ mV and $-31$ mV ($P = 0.054$, two-way ANOVA; voltages more positive than $-40$ mV were not considered to exclude possible contamination from the delayed rectifier $K^+$ current which activates positive to $-40$ mV, see Spaiardi et al., 2017; Fig. 7$C$,$D$). Therefore it is likely that the significantly smaller peak transient $\Delta C_m$ in $Caspr^{-/-}$ type I hair cells is due to the larger synaptic cleft, i.e., to the smaller $R_c$ and consequently lower amplification of the transient $\Delta C_m$ associated with K,L channel gating currents, and not from differences in the numbers or kinetics of K,L channels.

## Discussion

Vestibular type I hair cells release glutamate upon $Ca^{2+}$ inflow through voltage-gated $Ca_V1.3$ channels (see Mukhopadhyay & Pangrsic, 2022 for a recent review). Present results show that sustained neurotransmitter exocytosis is nearly abolished in 10 mM intracellular EGTA (Fig. 2$A$, $B$ and $G$). Because of its slow forward rate for $Ca^{2+}$ binding, EGTA does not capture $Ca^{2+}$ in very close proximity (tens of nm) of the open $Ca^{2+}$ channel, for which the fast $Ca^{2+}$ chelator BAPTA is required, whereas it chelates $Ca^{2+}$ entered and diffusing at a μm distance from the open $Ca^{2+}$ channels (Neher, 1998). The above results are therefore consistent with $Ca^{2+}$ channels and vesicle release sites being within a microdomain. This is in contrast with previous studies showing that EGTA did not affect (Dulon et al., 2009) or only partially blocked (~40%; Vincent et al., 2014) exocytosis in mouse type I hair cells, which was conversely fully blocked by BAPTA (both $Ca^{2+}$ chelators tested at 5 mM). It is likely that the difference is due to the mouse age, which in their studies ranged from postnatal day (P)4 to P9, while here it was between P17 and P19. Differences between neonatal and adult mouse vestibular type II hair cells have been

reported concerning the $Ca^{2+}$-dependence of exocytosis, which was linear in neonatal type II hair cells (Dulon et al., 2009), but high order in the adult (>P18) ones (Spaiardi et al., 2022). Several changes are known to occur at the ribbon synapse during hair cell maturation (e.g. $Ca^{2+}$ channel localization, ribbon anatomy; see Pangrsic et al., 2018 and Michanski et al., 2023 for a recent review), which may be responsible for the above different sensitivity to intracellular $Ca^{2+}$ buffers with age.

The large transient $\Delta C_m$ recorded in type I hair cells was nearly unaffected by 10 mM intracellular EGTA (Fig. 2$A$–$F$), conclusively demonstrating that it does not depend on intracellular $Ca^{2+}$. On the contrary, the transient $\Delta C_m$ correlated with the activation curve of $G_{K,L}$ (the dominant ionic conductance in type I hair cells) – Fig. 4$E$, indicating that it is likely to be generated by the intramembrane movement of the charges associated with the voltage-dependent gating of K,L channels, that are present at very high density (about 150 per μm$^2$ in rat type I hair cells; Chen & Eatock, 2000). A large transient $\Delta C_m$, unrelated to $Ca^{2+}$-dependent exocytosis, has been reported in adrenal chromaffin cells and was shown to be due to $Na^+$ channel gating charge movement associated with channel de/inactivation (Horrigan & Bookman, 1994), the magnitude and time course of which were like that seen in type-I vestibular hair cells. Such transient capacitance signal reflecting $Na^+$ channel-gating charge movement dibucaine could be cancelled by 200 μM dibucaine that blocks both $Na^+$ current ($I_{Na}$) and $Na^+$ channel-gating charge movement in squid axon (Gilly & Armstrong, 1980), but not by TTX, which blocks I Na but does not immobilize $Na^+$ channel gating charges. Clearly TEA and 4-AP, here added to the extracellular solution together with intracellular Cs to block $I_{K,L}$, do not immobilize the related gating charges. Finding a drug that immobilizes K,L channel-gates remains an interesting task, also given the recent identification of Kv 1.8 (Kcna10) channel subunits as responsible for carrying $I_{K,L}$ (Martin et al., 2023). A better knowledge of $K_v1.8$ properties is desirable given that *KCNA10* is expressed in the heart, aorta and kidney. Very recently a missense mutation of *KCNA10 h*as been involved in epinephrine provoked long QT syndrome with a familial history of sudden cardiac death (Huang et al., 2023).

The amplitude of the transient $\Delta C_m$ was significantly smaller in $Caspr^{-/-}$ type I hair cells than in control cells. Caspr is a core molecular component of septate junctions, which in vertebrates are found only in myelin paranodal contacts (Banerjee et al., 2006). These junctions restrict ion movement between the extracellular space confined in the insulated portion of the myelinated axons (internodes) and the extracellular space surrounding the nodes of Ranvier (Salzer, 2003). A similar purpose, confinement of $K^+$ ions, would therefore be served at the vestibular calyceal synaptic cleft. Additionally many

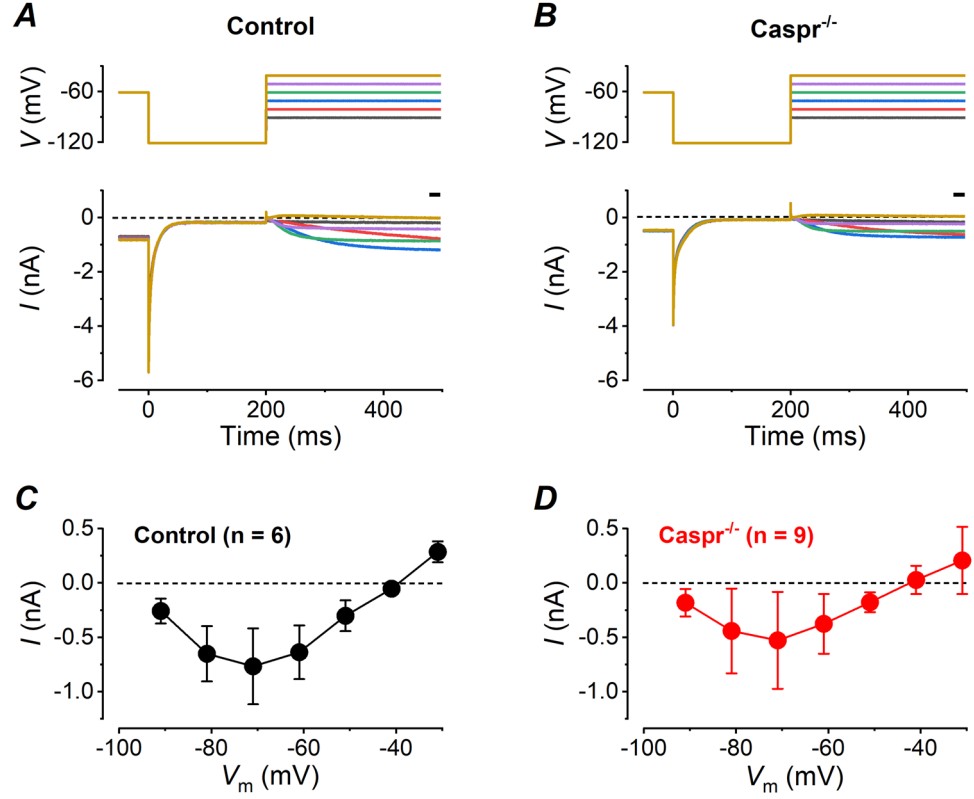

**Figure 7. $G_{K,L}$ in control and *Caspr$^{-/-}$* type I hair cells of mature mouse utricles**
*A*, *B*, representative macroscopic currents from a type I hair cell of a control and a *Caspr$^{-/-}$* mouse, respectively. Currents were recorded by using the CsGlutamate-based intracellular solution described in the Methods section. From a $V_{hold}$ of −61 mV the cell was first hyperpolarized to −121 mV, and then iteratively depolarized by stepping the voltage from −91 to −31 mV in 10 mV increments, as shown by the voltage protocols shown in the upper panels. *C*, *D*, average steady-state macroscopic current measured in relation to the black bars shown above the traces as a function of $V_m$ in control (*n* = 6) and *Caspr$^{-/-}$* (*n* = 9) type I hair cells, respectively.

papers have shown that synaptic ectodomains and the proteins aggregated there also modulate the expression and function of nearby ion channels (Brockhaus et al., 2018; Wierda et al., 2020). For instance the septate junction protein Mesh plays an essential role in the physiological maturation and function of the Drosophila Malpighian tubule epithelium that is required for normal transepithelial fluid, ion transport and paracellular permeability (Jonusaite et al., 2020). Indeed Caspr is required for the recruitment or retention of KCNQ4 K$^+$ channels at the vestibular calyces (Sousa et al., 2009). However we found that the lack of Caspr did not affect the expression or properties of $G_{K,L}$. Therefore the smaller $\Delta C_m$ found in *Caspr$^{-/-}$* type I hair cells is likely the consequence of the smaller $R_c$, given the larger distance between the hair cell basolateral membrane and the inner face of the calyx terminal compared with its wild-type counterpart (Sousa et al., 2009). However it cannot be excluded that this smaller $\Delta C_m$ could also be produced by the lack of the septate junction and/or by the proteins aggregated there, which could be not expressed, washed away or misplaced in the lack of this junction. These

proteins could be charged and dragged in some way by the $G_{K,L}$ gate, amplifying the gating transient when they are present: on the basis of the literature cited above, these proteins could link pre- and post-synaptic elements, contributing to the accelerated synaptic transmission. It is excluded that the lack of these proteins decreases $\Delta C_m$ by slowing down the kinetics of the gating charge movement, and/or by decreasing the number of K,L channels, since this would affect the $I_{K,L}$ activation/deactivation kinetics and/or its amplitude, respectively, and this is not the case (Fig. 7).

As a final consideration, it is worth considering that $R_c$ can only be estimated; that is, it cannot be measured directly. Recently, Cohen et al. (2020) estimated a resistivity of the extracellular solution of about 550 $\Omega$ cm at the paranode, i.e., about five times larger than that typically estimated for the resistivity of the extracellular solution at the calyceal synaptic cleft (100 $\Omega$ cm) (Govindaraju et al., 2023). Given its resemblance to the paranodal septate junction, even the resistivity of the extracellular solution at the calyceal synaptic cleft might be significantly higher than assumed, and so would $R_c$.

Indeed, despite piercing the calyx, recordings from the enclosed type I hair cell show that an outward $K^+$ current through $G_{K,L}$ of a few nA increased $K^+$ in the (residual) calyceal synaptic cleft of a few tens of mM (Contini et al., 2012; Spaiardi et al., 2017), as also shown by the following simple calculation. A $G_{K,L}$ current of amplitude $i_K$ corresponds to an increase in the cleft concentration of $n$ moles/s given by:

$$n = \frac{i_K}{e N_o C_v}$$

where $e$ is the elementary charge, $N_o$ is the Avogadro number and $Vol_{cl}$ is the volume of the cleft (Table 1), which is

$$Vol_{cl} = \frac{4}{3} \pi \left[ (r_h + s_c)^3 - r_h^3 \right] f \approx 4\pi r_h^2 s_c f$$

In the $Vol_{cl}$ expression the terms in $s_c^2$ and $s_c^3$ have been neglected because they are negligible in respect to $r_h^2$. Therefore:

$$n = \frac{i_K}{4\pi e N_o r_h^2 s_c f}$$

In round numbers, an $i_K$ of $\approx 400$ pA causes the cleft $K^+$ concentration to rise at a rate of $n = 3500$ mM/s in the absence of any transport (pumps, channels or exchangers) in the hair cell or in the calyx plasma membrane that removes cleft $K^+$. Assuming that the latter transports are not simultaneously activated to the $i_K$ onset, the resting cleft $K^+$ concentration, $[K^+]_o \approx 5$ mM, would rise to an amount of $\Delta[K^+]_o \approx 4$ mM within 1 ms, causing a calyx depolarization of $\Delta V \approx 15$ mV due to its open $K^+$-permeable channels (mainly $K_v 7$ ones) (Spitzmaul et al., 2013), according to:

$$\Delta V = \frac{RT}{F} ln \left( \frac{\left[ K^+ \right]_o + \Delta \left[ K^+ \right]_o}{\left[ K^+ \right]_o} \right)$$

that is enough to trigger an action potential in the calyx. This large $K^+$ efflux is not expected to cause any charge accumulation in the cleft, because of the rapid redistribution due to the strong electrostatic interaction between $K^+$ and the other ions (mainly $Na^+$ and $Cl^-$) present there. This calculation, although very basic, is in agreement with a very sophisticated model of the non-quantal transmission between type I hair cell and the calyx (Govindaraju et al., 2023), describing the kinetics of $K^+$ accumulation in the cleft.

Thus it seems reasonable to assume that, with an undamaged calyx, the resistivity of the extracellular solution in the synaptic cleft is higher than generally assumed, consistent with a residual $R_c$ of about 40 MΩ in a pierced calyx (resulting from a ∼ 15 mV depolarization for an IK,L of ∼ 400 pA).

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

## Additional information

### Data availability statement

Data will be made available upon reasonable request.

### Competing interests

The authors declare that they have no competing interests.

### Author contributions

P.S., S.M. and S.L.J. designed the study. P.S. and S.L.J. performed the experiments. P.S., S.M., G.R. and S.L.J. analysed and interpreted the data. P.S., S.M., G.R., R.G. and S.L.J. critically revised and approved the final version of the manuscript submitted for publication.

### Funding

This work was supported by grants from THE EUROPEAN UNION – NEXT GENERATION EU, PNRR M4.C2.1.1 – 20228AAJRL – Signal transmission at the mammalian vestibular hair cell synapses – CUP: F53D23005930006 to Sergio Masetto and Giorgio Rispoli. THE EUROPEAN UNION – NEXT GENERATION EU, Missione 4 Componente 1 – 2022KNFAYR—BIO-SMILE: BIOmedical restoration of Symmetrical Motor control in unilateral paraLysEs due to nerve lesions – CUP: F53D23006030001. BBSRC (BB/X000567/1) and RNID (G106) to Stuart L Johnson.

### Acknowledgements

Caspr mice were a kind gift from Prof. Elior Peles, Weitzmann Institute of Science (Rehovot, Israel).

Open access publishing facilitated by Universita di Pavia, as part of the Wiley - CRUI-CARE agreement.

### Keywords

cell membrane capacitance, exocytosis, gating current, hair cell, vestibular

## Supporting information

Additional supporting information can be found online in the Supporting Information section at the end of the HTML view of the article. Supporting information files available:

**Peer Review History**

