## [Peer Review History · The Journal of Physiology]

Transient capacitance changes recorded from vestibular type I hair cells are produced by $G_{K,L}$ gating and do not involve neurotransmitter exocytosis

Paolo Spaiardi, Roberta Giunta, Giorgio Rispoli, Sergio Masetto, and Stuart Leigh Johnson

DOI: 10.1113/JP288645

Corresponding author(s): Paolo Spaiardi (paolo.spaiardi@unipv.it)

The following individual(s) involved in review of this submission have agreed to reveal their identity: Ruth Anne Anne Eatock (Referee #1)

Review Timeline:

Submission Date:	11-Feb-2025
Editorial Decision:	08-Apr-2025
Revision Received:	14-May-2025
Editorial Decision:	19-Jun-2025
Revision Received:	23-Jun-2025
Accepted:	09-Jul-2025

Senior Editor: Nathan Schoppa

Reviewing Editor: Tina Pangršič

Transaction Report:

Dear Dr Spaiardi,

Re: JP-RP-2025-288645 "Transient capacitance changes recorded from vestibular type-I hair cells are produced by $G_{K,L}$ gating and do not involve neurotransmitter exocytosis" by Paolo Spaiardi, Roberta Giunta, Giorgio Rispoli, Sergio Masetto, and Stuart Leigh Johnson

Thank you for submitting your manuscript to The Journal of Physiology. It has been assessed by a Reviewing Editor and by 2 expert referees and we are pleased to tell you that it is acceptable for publication following satisfactory revision.

LANGUAGE EDITING AND SUPPORT FOR PUBLICATION: If you would like help with English language editing, or other article preparation support, Wiley Editing Services offers expert help, including English Language Editing, as well as translation, manuscript formatting, and figure formatting at www.wileyauthors.com/eoo/preparation. You can also find resources for Preparing Your Article for general guidance about writing and preparing your manuscript at www.wileyauthors.com/eoo/prepresources.

REVISION CHECKLIST:

We look forward to receiving your revised submission.

Yours sincerely,

Nathan Schoppa
Senior Editor
The Journal of Physiology

REQUIRED ITEMS

- You must start the Methods section with a paragraph headed Ethical approval (https://jp.msubmit.net/cgi-bin/main.plex?form_type=display_requirements#methods).

Research must comply with The Journal's policies regarding animal experiments (<https://physoc.onlinelibrary.wiley.com/hub/animal-experiments>) and adherence to these policies must be stated in the manuscript.

Authors should confirm in their Methods section that their experiments were carried out according to the guidelines laid down by their institution's animal welfare committee, including an ethics approval reference number. The Methods section must contain a statement about access to food, water and housing, details of the anaesthetic regime: anaesthetic used, dose and route of administration, and method of killing the experimental animals.

- Your manuscript must include a complete Additional Information section, including competing interests; funding; author contributions and acknowledgements.

- Please upload separate high-quality figure files via the submission form.

- Please ensure that the Article File you upload is a Word file.

- A Data Availability Statement is required for all papers reporting original data. This must be in the Additional Information section of the manuscript itself. It must have the paragraph heading 'Data Availability Statement'. All data supporting the results in the paper must be either: in the paper itself; uploaded as Supporting Information for Online Publication; or archived in an appropriate public repository. The statement needs to describe the availability or the absence of shared data. Authors must include in their statement: a link to the repository they have used, or a statement that it is available as Supporting Information; reference the data in the appropriate sections(s) of their manuscript; and cite the data they have shared in the References section. Whenever possible, the scripts and other artefacts used to generate the analyses presented in the paper should also be publicly archived. If sharing data compromises ethical standards or legal requirements then authors are not expected to share it, but must note this in their statement. For more information, see our Statistics Policy.

- You must start the Methods section with a paragraph headed Ethical approval (https://jp.msubmit.net/cgi-bin/main.plex?form_type=display_requirements#methods).

Research must comply with The Journal's policies regarding animal experiments (<https://physoc.onlinelibrary.wiley.com/hub/animal-experiments>) and adherence to these policies must be stated in the manuscript.

Authors should confirm in their Methods section that their experiments were carried out according to the guidelines laid down by their institution's animal welfare committee, including an ethics approval reference number. The Methods section must contain a statement about access to food, water and housing, details of the anaesthetic regime: anaesthetic used, dose and route of administration, and method of killing the experimental animals.

- Your manuscript must include a complete Additional Information section, including competing interests; funding; author contributions and acknowledgements.

- The Journal of Physiology funds authors of provisionally accepted papers to use the premium BioRender site to create high resolution schematic figures. Follow this link and enter your details and the manuscript number to create and download figures. Upload these as the figure files for your revised submission. If you choose not to take up this offer, we require figures to be of similar quality and resolution. If you are opting out of this service to authors, state this in the Comments section on the Detailed Information page of the submission form. The link provided should only be used for the purposes of this submission. Authors will be charged for figures created on this premium BioRender account if they are not related to this manuscript submission.

- Please upload separate high-quality figure files via the submission form.

- Please ensure that the Article File you upload is a Word file.

- Papers must comply with the Statistics Policy: https://jp.msubmit.net/cgi-bin/main.plex?form_type=display_requirements#statistics.

In summary:

- If $n \leq 30$, all data points must be plotted in the figure in a way that reveals their range and distribution. A bar graph with data points overlaid, a box and whisker plot or a violin plot (preferably with data points included) are acceptable formats.

- If $n > 30$, then the entire raw dataset must be made available either as supporting information, or hosted on a not-for-profit repository, e.g. FigShare, with access details provided in the manuscript.

- 'n' clearly defined (e.g. x cells from y slices in z animals) in the Methods. Authors should be mindful of pseudoreplication.

- All relevant 'n' values must be clearly stated in the main text, figures and tables.

- The most appropriate summary statistic (e.g. mean or median and standard deviation) must be used. Standard Error of the Mean (SEM) alone is not permitted.

- Exact p values must be stated. Authors must not use 'greater than' or 'less than'. Exact p values must be stated to three significant figures even when 'no statistical significance' is claimed.

- A Data Availability Statement is required for all papers reporting original data. This must be in the Additional Information section of the manuscript itself. It must have the paragraph heading 'Data Availability Statement'. All data supporting the results in the paper must be either: in the paper itself; uploaded as Supporting Information for Online Publication; or archived in an appropriate public repository. The statement needs to describe the availability or the absence of shared data. Authors must include in their statement: a link to the repository they have used, or a statement that it is available as Supporting Information; reference the data in the appropriate sections(s) of their manuscript; and cite the data they have shared in the References section. Whenever possible, the scripts and other artefacts used to generate the analyses presented in the paper should also be publicly archived. If sharing data compromises ethical standards or legal requirements then authors are not expected to share it, but must note this in their statement. For more information, see our Statistics Policy.

- Please include an Abstract Figure file, as well as the Figure Legend text within the main article file. The Abstract Figure is a piece of artwork designed to give readers an immediate understanding of the research and should summarise the main conclusions. If possible, the image should be easily 'readable' from left to right or top to bottom. It should show the physiological relevance of the manuscript so readers can assess the importance and content of its findings. Abstract Figures should not merely recapitulate other figures in the manuscript. Please try to keep the diagram as simple as possible and

without superfluous information that may distract from the main conclusion(s). Abstract Figures must be provided by authors no later than the revised manuscript stage and should be uploaded as a separate file during online submission labelled as File Type 'Abstract Figure'. Please also ensure that you include the figure legend in the main article file. All Abstract Figures should be created using BioRender. Authors should use The Journal's premium BioRender account to export high-resolution images. Details on how to use and access the premium account are included as part of this email.

EDITOR COMMENTS

Reviewing Editor:

Comments for Authors to ensure the paper complies with the Statistics Policy:

To comply with the journal statistics policy, please, use SD instead of SEM, provide precise p values (rather than > or <) and include data points on the graphs when $n < 30$ per averaged point.

Comments to the Author:

The MS entitled "Transient capacitance changes recorded from vestibular type-I hair cells are produced by GK,L gating and do not involve neurotransmitter exocytosis" was now seen by two reviewers who both very much appreciate the relevance of the findings, but also both provide comments that should be addressed, including the concerns regarding the channel blockers. Please, address all comments carefully in the revised version of the manuscript. In addition, to comply with the journal statistics policy, please, use SD instead of SEM, provide precise p values (rather than > or <) and include individual data points in the graphs whenever $n < 30$ (per averaged point).

We are looking forward to receiving a revised version of your MS.

Senior Editor:

Comments for Authors to ensure the paper complies with the Statistics Policy:

Please see comments from the Reviewing Editor on statistics. Some concerns were raised.

Comments to the Author:

Thank you for submitting your manuscript to Journal of Physiology. It has been generally favorably evaluated by two expert reviewers and the reviewing editor, who felt that the study is potentially quite impactful. Both reviewers and the reviewing editors have raised concerns all of which will need to be addressed in a revised manuscript that will be re-reviewed. The changes, including those added to address the point from reviewer 2 about the pharmacology, should not require additional experiments.

REFEREE COMMENTS

Referee #1:

Here Spaiardi and co-authors significantly advance our understanding of afferent transmission at the type I-calyx synapse, an extraordinary synapse with multiple transmission components. This work probes how salient features of the synapse - extensive calyceal wrapping of the hair cell plus dense hair cell expression of low-voltage-activated potassium channels (G-K,L) - interact to affect a key measure of glutamate release (exocytosis). With several manipulations, they show that an unusual transient capacitive current is likely the gating current of G-K,L channels rather than an increase in membrane surface area caused by vesicle fusion with the plasma membrane.

General comments

Spaiardi, Johnson and colleagues here add to their systematic analysis of glutamate transmission mechanisms at hair cell afferent synapses of the mammalian vestibular inner ear, specifically the mouse utricular macula. The synapse between type I hair cell and calyceal afferent terminal, which is unique to amniotes, has been shown variously to use vesicular glutamate release (exocytosis) and an unusual ion-channel based form of non-vesicular transmission. While progress has been made in unraveling afferent transmission at this specialized synapse, many questions remain.

In this paper, Spaiardi, Johnson et al. use the capacitance method with whole-cell recordings from hair cells to register

exocytosis by the increases in C_m following vesicle fusion. A previous report (Spaiardi et al. 2022) showed that type I cells have a large transient capacitive current lacking the expected dependence on voltage-gated calcium current. Here they pursue the nature of the transient current and find that it comprises gating currents of the large number of hair cell G-K,L channels, based on 1) a similar voltage range to G-K,L activation; 2) insensitivity to elevated intracellular Ca^{2+} buffering (unlike exocytosis); and 3) reduction in Caspr-null mice, which Sousa et al. (2009) showed have an expanded calyceal synaptic cleft and disrupted expression/localization of Kv channels (though not G-K,L channels, for which they did not have an antibody).

Detailed comments

Introduction

3. Clarify for readers that Caspr is expressed by the postsynaptic afferent membrane, and describe its known effects on KCNQ4 distribution there. With the ms. focused on the type I side of the synapse, readers might think that Caspr is expressed by the hair cells, rather than the afferent as emphasized by Sousa et al. This could be clarified by adding Caspr to Fig. 1. (Note that Caspr does show up in some RNAseq data (see gEAR) from hair cells and supporting cells, especially at young ages, such that the constitutive KO might have wider developmental effects than appreciated.)

Methods

4. "Piercing" sounds like what a sharp electrode would do, not a patch electrode. Do you perhaps clear the outer-face membrane and pierce the tightly adhered inner-face membrane? If so, is the experience different in the Caspr-null mouse, where the calyx inner-face membrane is less adherent?

5. For the sake of reproducibility, provide information about the Caspr-null mice - are these a constitutive knockout and of the same pedigree as Kachar and colleagues used in 2009 (also not identified)? If no citation is available, please provide the genetic strategy used by the lab that provided the animals.

6. Indicate how many animals provided the 6 control cells and 11 null cells. Also indicate whether controls include hets and WT; if there is a mix, I am guessing there are too few to indicate one way or the other whether there is a het effect relative to WT ?

7. The high Cs^+ permeability of G-K,L (~ 0.3 relative to K^+ permeability) is characterized in Rusch's 1996 paper.

8. Equations

- Please add a table of parameters etc. and citations for assumptions.
- Avoid re-using labels for very different quantities ('V' is Voltage in Results and Volume in Discussion)
- Eq. 5: arctan, not arctang

Results

9. p. 16 Eq 1 : f = fraction of basolateral hair cell membrane (below tight junctions) enwrapped by calyx

10. Text following Eq. 5: Physiologically, type I R_m is much lower than 1 gigohm (much closer to R_s of 10 Megohms) - does this affect the Santos-Sacchi 2004-type simulations of C_m artifact? (touched on later in the paragraph in the context of

artifactual Cm readings as Rm and Rs approach each other)

Discussion

11. p. 22 - See also Michanski et al. 2023 for developmental/aging changes in the proximity of ribbons to plasma membrane, and in ribbon clustering.

12. p. 23

- Add citation(s) for the "typically estimated" resistivity of the extracellular solution at the calyceal synaptic cleft (100 ohm-cm)

- Number the equations as for previous equations

- Correct subscript in the text between 2nd and 3rd equation: "sc3" should be Sc3"

- "in round numbers"

- "resting cleft K+ concentration"

- Cite source(s) for Kv7 channels in calyx

13. Vincent, Dulon and colleagues have reported exocytosis measurements for mouse utricular type I hair cells at earlier stages than studied here. The current paper predicts that the transient capacitance would not be present until G-K,L emerges, mostly over the first postnatal week - is it worth checking whether that is available in the earlier-stage data?

14. The arguments of the last paragraph ("This is consistent with a residual Rc of about 40 Megohms, as by the following calculation") could be reframed to end on the answer promised. Ballpark calculations estimate ~15 mV depolarization for 400 pA I-K,L in the case of a pierced calyx, which does indeed correspond to "pierced-calyx" Rc ~ 40 Megohms, but perhaps could be made explicit to close out the thought.

Figure Legends

15. Figure 7 - Are C,D steady-state currents? n's should be in figure legend.

- Associated text above figure is ambiguous as written, change to: "i.e., to the smaller Rc and consequently lower amplification of the transient Cm...".

Referee #2:

This study provides valuable insights into the role of GK,L channels in regulating transient capacitance (ΔC_m) in type I hair cells. By demonstrating that ΔC_m transients are significantly influenced by the gating of IK,L, the authors offer new perspectives on the physiological mechanisms underlying hair cell function and synaptic signaling. The use of capacitance measurements to track gating charge movements is an innovative and effective approach to investigate the activity of this channel.

Impact on the area of research:

The study significantly advances our understanding of ion channel function in hair cells, particularly within the context of

synaptic signaling at the type I hair cell-calyx synapse. By focusing on IK,L channels, and their gating behavior, the research opens new avenues for exploring how synaptic transmission in vestibular hair cells is regulated at the ion channel level. This is particularly relevant in light of recent discoveries related to fast resistive coupling, which relies on ion channels in both type I hair cells and the calyx. These findings could have broader implications for understanding hair cell physiology and may pave the way for therapeutic targeting of ion channels in disorders affecting sensory cells.

Insight into physiological mechanisms in this field:

The study sheds light on the role of IK,L in regulating capacitance transients and synaptic activity in hair cells. By examining the gating of IK,L channels and their contribution to ΔC_m , the authors provide crucial insight into the mechanisms that govern synaptic function in type I hair cells. The use of holding potentials to assess channel gating provides a valuable experimental framework to understand how ionic conditions and channel dynamics influence cellular processes. Furthermore, the study's comparison of ΔC_m transients in wild-type and Caspr^{-/-} mice suggests that changes in the synaptic cleft geometry can influence capacitance changes, thereby providing a clearer picture of the factors that modulate synaptic efficiency.

Originality of the research:

The originality of this research lies in its approach of using capacitance measurements as a readout of IK,L gating. This technique is relatively underutilized in the study of synaptic ion channels and provides a novel way to probe channel dynamics. By using capacitance transients to explore the gating behavior of IK,L, the study offers a fresh perspective on the role of potassium channels in hair cell function and synaptic transmission. The combination of this method with genetic models (wild-type vs. Caspr^{-/-}) further strengthens the originality of the study by providing a comprehensive approach to understanding the interaction between channel function and synaptic architecture.

Study design and robustness of the experimental data:

The experimental design is generally robust, with careful measurement of ΔC_m transients at various holding potentials, which strengthens the hypothesis that IK,L gating plays a key role in regulating capacitance changes. The comparison between wild-type and Caspr^{-/-} mice is an effective way to isolate the effects of changes in the synaptic cleft geometry on capacitance transients. However, the use of pharmacological blockers (4AP and TEA) to isolate the contribution of IK,L raises potential concerns regarding off-target effects, which could complicate the interpretation of ΔC_m measurements. For example, blockers may influence other ion channels or disrupt cellular processes, potentially confounding the results.

Validity of the conclusions:

While the study's conclusions are largely valid, particularly with regard to the role of IK,L gating in regulating ΔC_m , the use of blockers introduces uncertainties that could affect the validity of the findings. The potential for off-target effects of 4AP and TEA raises concerns about whether the observed ΔC_m changes are directly attributable to the gating of IK,L channels or reflect broader disruptions in cellular processes. If the blockers effectively block IK,L, the gating charge contributions of this channel should not be detectable in the capacitance transients, which contradicts the authors' claim that transient ΔC_m can serve as a reliable readout of IK,L gating. That said, the authors provide strong evidence supporting their hypothesis, and the overall conclusions contribute to our understanding of the role of GK,L in hair cell function. Nevertheless, addressing the limitations of the pharmacological approach would strengthen the validity of the study's conclusions.

Please see detailed comments in the attached document.

END OF COMMENTS

General Comments:

This study provides valuable insights into the role of $G_{K,L}$ channels in regulating transient capacitance (ΔC_m) in type I hair cells. The authors demonstrate that ΔC_m transients are significantly influenced by the gating of $I_{K,L}$, and the use of capacitance measurements to track gating charge movements is an effective way to probe the activity of this channel. The experimental approach, including the measurement of ΔC_m transients at various holding potentials (-81 mV and -131 mV), supports the hypothesis that $I_{K,L}$ gating is a key mechanism underlying the capacitance changes observed.

However, a key aspect of the experimental design is the use of blockers to isolate the contribution of $I_{K,L}$ channels. While this is a standard technique, it introduces potential confounds that should be considered. The blockers used to inhibit $I_{K,L}$ (4AP and TEA) may have off-target effects, either by influencing other ion channels or by altering cellular processes such as membrane potential and synaptic activity. Such effects could complicate the interpretation of ΔC_m measurements, as changes in capacitance could reflect a broader disruption of ion channel dynamics rather than a specific gating of $I_{K,L}$. If $G_{K,L}$ is indeed blocked, its gating charges should not contribute to ΔC_m , contradicting the authors' claim that transient ΔC_m can serve as a readout of $G_{K,L}$ gating.

Moreover, the direct modulation of $I_{K,L}$ using blockers may also impact synaptic function. Given that $I_{K,L}$ is involved in the synaptic signaling at the type I hair cell–calyx synapse, blocking these channels could alter the mechanisms the study aims to investigate, introducing additional complexities in the interpretation of the results.

That said, the study's approach is still highly informative. The comparison of ΔC_m amplitudes in wild-type and *Caspr*^{-/-} mice provides compelling evidence that the size of the capacitance transient is affected by changes in the synaptic cleft, with *Caspr*^{-/-} mice exhibiting smaller capacitance changes due to alterations in the synaptic cleft geometry. The finding that the expression or kinetics of $I_{K,L}$ were not significantly altered in *Caspr*^{-/-} cells strengthens the argument that the differences in ΔC_m are due to changes in R_c rather than in channel function or expression.

In conclusion, while the use of blockers introduces certain limitations, the data presented in this study significantly advance our understanding of how $I_{K,L}$ gating regulates ΔC_m at the type I hair cell–calyx synapse. Moving forward, it would be beneficial to explore more refined techniques that can isolate the contribution of $I_{K,L}$ without introducing potential confounds associated with pharmacological blockers.

Specific Comments:

- I noticed some variation in the terminology used to describe hair cell types. The standard convention in scientific literature is to write "**type I hair cells**" and "**type II hair cells**" without a hyphen. I recommend maintaining consistency throughout the manuscript.

- There are a few typographical errors in the manuscript. For example, in the sentence:
“The above findings suggest that the large transient ΔC_m recorded from type-I hair cells does not depend upon Ca^{2+} entry though voltage-gated $CaV1.3 Ca^{2+}$ channels.”
 "though" should be corrected to "through."
- I noticed some inconsistencies in the capitalization of the first letter in figure legends. I recommend maintaining consistency, as it is common practice to capitalize the first letter of each sentence.
- **Figure 2:** I suggest specifying that 4-AP and TEA were added. Additionally, it would be helpful to clarify that the thick line represents the sine wave. Regarding Figure 2B, I have a question: Why does the current appear inward at 10 mM EGTA?
- **Figure 3 (Panels E, F):** The text states:
“E, F: average differential trace obtained by subtracting ΔC_m recorded at +10 mV from that recorded at -21 mV.”
 However, this should be corrected to:
“E, F: average differential trace obtained by subtracting ΔC_m recorded at +19 mV from that recorded at -21 mV.”
- **Figure 4 (Panel C):**
 The legend states:
“C: average ΔC_m recorded upon repolarization to -81 mV or -131 mV following voltage steps ranging from -101 mV to -1 mV.”
 However, the figure itself specifies -81 mV. Could you clarify which value is correct?
- **Figure 4 (Panel D):**
 The legend states:
“D: average ΔC_m recorded upon repolarization to -131 mV following voltage steps ranging from -151 mV to -1 mV.”
 However, earlier in the text, it is mentioned that the voltage range is -141 mV to -1 mV. Could you confirm which range is correct?
- **In the section ‘Does the calyx play a role in the large transient ΔC_m ?’**
 The sentence:
“Indeed, even the most depolarized voltage steps elicited outward (Ca^{2+}) current ≤ 100 pA in either cell types (Fig. 1A and B), meaning that R_m was at all voltages ≥ 1 G Ω , i.e. much larger than the typical R_s values.”
 refers to Figure 1A and B, but these panels are cartoons, and the figure actually displays calcium currents. I suggest clarifying the reference to avoid confusion.

Transient capacitance changes recorded from vestibular type I hair cells are produced by GK,L gating and do not involve neurotransmitter exocytosis

We thank the Reviewers for their constructive and very helpful comments, which have helped us to strengthen the manuscript. Replies to the comments raised are detailed below.

Reviewing Editor and Senior Editor:

Comments to the Author:

The MS entitled "Transient capacitance changes recorded from vestibular type-I hair cells are produced by GK,L gating and do not involve neurotransmitter exocytosis" was now seen by two reviewers who both very much appreciate the relevance of the findings, but also both provide comments that should be addressed, including the concerns regarding the channel blockers. Please, address all comments carefully in the revised version of the manuscript. In addition, to comply with the journal statistics policy, please, use SD instead of SEM, provide precise p values (rather than > or <) and include individual data points in the graphs whenever $n < 30$ (per averaged point).

We thank the Editor and Senior Editor for their comments on our manuscript. In terms of the channel blockers we have used we have justified the reasons for this in our first reply to Referee 2

We have changed the statistics to include the precise P values. In a few cases we have left $P < 0.0001$ since we could not obtain the exact value from the statistics program we used.

Referee #1:

We thank the Reviewer for their favorable opinion of our manuscript.

Detailed comments

Introduction

3. Clarify for readers that Caspr is expressed by the postsynaptic afferent membrane, and describe its known effects on KCNQ4 distribution there. With the ms. focused on the type I side of the synapse, readers might think that Caspr is expressed by the hair cells, rather than the afferent as emphasized by Sousa et al. This could be clarified by adding Caspr to Fig. 1. (Note that Caspr does show up in some RNAseq data (see gEAR) from hair cells and supporting cells, especially at young ages, such that the constitutive KO might have wider developmental effects than appreciated.)

Thank you for the suggestion, we have now added Caspr to Fig. 1. Although we mentioned in the text (pag.16) of the submitted ms that Caspr is required for the recruitment or retention of KCNQ4 K⁺ channels at the vestibular calyces (Sousa et al., 2009), this information could indeed be overlooked given the focus on hair cells.

Methods

4. "Piercing" sounds like what a sharp electrode would do, not a patch electrode. Do you perhaps clear the outer-face membrane and pierce the tightly adhered inner-face membrane? If so, is the experience different in the Caspr-null mouse, where the calyx inner-face membrane is less adherent?

As previously reported (e.g., Spaiardi et al., 2017), we used a patch pipette to remove the tissue debris, above the targeted hair cell prior to sealing onto it. It is likely that both the outer and inner calyx membrane below the patch pipette were aspirated because of the negative pressure used to seal and to rupture the hair cell membrane. We observed no significant differences between WT and KO, possibly because the above procedure was rather variable among experiments. This is now specified in the Methods (Page 5, line 133).

5. For the sake of reproducibility, provide information about the Caspr-null mice - are these a constitutive knockout and of the same pedigree as Kachar and colleagues used in 2009 (also not identified)? If no citation is available, please provide the genetic strategy used by the lab that provided the animals.

We mentioned in the text (Page 4, line 115) that Caspr mice were a kind gift from Prof. Elijor Peles, Weizmann Institute of Science (Rehovot, Israel), and have now provided the following relevant reference:

Gollan, L., Salomon, D., Salzer, J. L., & Peles, E. (2003). Caspr regulates the processing of contactin and inhibits its binding to neurofascin. *The Journal of cell biology*, 163(6), 1213–1218. <https://doi.org/10.1083/jcb.200309147>

6. Indicate how many animals provided the 6 control cells and 11 null cells. Also indicate whether controls include hets and WT; if there is a mix, I am guessing there are too few to indicate one way or the other whether there is a het effect relative to WT ?

We have now indicated in the legend of Figure 6 the number of animals from which control and Caspr^{-/-} cells were obtained. Caspr^{-/-} cells (n=11) were from 4 KO-mice. Control cells were from 1 Caspr Het mouse (n=2) and two wild type (n=4). No obvious differences were seen between WT and Het mice.

7. The high Cs⁺ permeability of G-K,L (~0.3 relative to K⁺ permeability) is characterized in Rusch's 1996 paper.

We have now added this citation (Page 5, line 143).

8. Equations

- Please add a table of parameters etc. and citations for assumptions.

Done, thank you.

- Avoid re-using labels for very different quantities ('V' is Voltage in Results and Volume in Discussion)

We replaced the cleft volume V_c with C_v throughout the text. Thank you.

- Eq. 5: arctan, not arctang

Done, thank you.

Results

9. p. 16 Eq 1 : f = fraction of basolateral hair cell membrane (below tight junctions) enwrapped by calyx

We modified the text according to the referee's suggestion.

10. Text following Eq. 5: Physiologically, type I R_m is much lower than 1 gigohm (much closer to R_s of 10 Megohms) - does this affect the Santos-Sacchi 2004-type simulations of C_m artifact? (touched on later in the paragraph in the context of artifactual C_m readings as R_m and R_s approach each other)

The Santos-Sacchi "e C_m " method is highly effective in measuring changes in C_m while avoiding interference from variations in either R_s or R_m . Simulations using electronic circuits (see their Fig. 5) demonstrated that C_m measurements remained unaffected even when R_m was reduced from 80 times to just 5 times the value of R_s . However, in real cell recordings (where substantial noise is present) and especially when R_m approaches the value of R_s , even this method may unfortunately produce capacitive artifacts.

Discussion

11. p. 22 - See also Michanski et al. 2023 for developmental/aging changes in the proximity of ribbons to plasma membrane, and in ribbon clustering.

We implemented the text (Page 15, line 423) and References accordingly.

12. p. 23

- Add citation(s) for the "typically estimated" resistivity of the extracellular solution at the calyceal synaptic cleft (100 ohm-cm)

We added the relevant citation (Govindaraju et al., 2023) (Page 16, line 471)

- Number the equations as for previous equations

Done, thank you.

- Correct subscript in the text between 2nd and 3rd equation: "sc3" should be Sc3"

Done, thank you.

- "in round numbers"

Done, thank you.

- "resting cleft K+ concentration"

Done, thank you.

- Cite source(s) for Kv7 channels in calyx

We added Spitzmaul et al., 2013 reference. (Page 17, line 490)

13. Vincent, Dulon and colleagues have reported exocytosis measurements for mouse utricular type I hair cells at earlier stages than studied here. The current paper predicts that the transient capacitance would not be present until G-K,L emerges, mostly over the first postnatal week - is it worth checking whether that is available in the earlier-stage data?

Yes. Indeed, we have preliminary data showing that neonatal (P4) mice show a smaller transient, which might be due to a smaller GK,L and/or calyx (wrapping).

14. The arguments of the last paragraph ("This is consistent with a residual Rc of about 40 Megohms, as by the following calculation") could be reframed to end on the answer

promised. Ballpark calculations estimate ~15 mV depolarization for 400 pA I-K,L in the case of a pierced calyx, which does indeed correspond to "pierced-calyx" $R_c \sim 40$ Megohms, but perhaps could be made explicit to close out the thought.

We thank the reviewer for this valuable suggestion, which improves the clarity and narrative of the conclusion. We have modified the final paragraph accordingly (Page 17, line 497).

Figure Legends

15. Figure 7 - Are C,D steady-state currents? n's should be in figure legend.

Yes, as shown by the small horizontal bars above the current traces. This is now specified in the legend. We have now added the n's values in the legend, thank you.

- Associated text above figure is ambiguous as written, change to: "i.e., to the smaller R_c and consequently lower amplification of the transient C_m ...".

Done, thank you.

Referee #2

General Comments:

This study provides valuable insights into the role of GK,L channels in regulating transient capacitance (ΔC_m) in type I hair cells. The authors demonstrate that ΔC_m transients are significantly influenced by the gating of IK,L, and the use of capacitance measurements to track gating charge movements is an effective way to probe the activity of this channel. The experimental approach, including the measurement of ΔC_m transients at various holding potentials (-81 mV and -131 mV), supports the hypothesis that IK,L gating is a key mechanism underlying the capacitance changes observed. However, a key aspect of the experimental design is the use of blockers to isolate the contribution of IK,L channels. While this is a standard technique, it introduces potential confounds that should be considered. The blockers used to inhibit IK,L (4AP and TEA) may have off-target effects, either by influencing other ion channels or by altering cellular processes such as membrane potential and synaptic activity. Such effects could complicate the interpretation of ΔC_m measurements, as changes in capacitance could reflect a broader disruption of ion channel dynamics rather than a specific gating of IK,L.

We thank the reviewer for these valuable comments. However, it's important to remember that the off-target effects of the blockers used to block $I_{K,L}$ should have a minimal impact on other conductances, given that $I_{K,L}$ is by far the dominant K^+ conductance in type-I hair cells. Similarly, their effect on the membrane potential should also be limited, considering that these experiments were performed under voltage clamp conditions. Furthermore, we must emphasize that, technically, there is no alternative to the use of blockers to establish the relationship between the calcium current (I_{Ca}) and exocytosis, as measured by membrane capacitance. Finally, we previously demonstrated (Spaiardi et al., 2020) that in identical experimental conditions, the transient ΔC_m was not present in type-II hair cells, which express several K^+ conductances but not $G_{K,L}$. Therefore, we are rather confident that the transient ΔC_m reflects $G_{K,L}$ gating. The blockers we have used (intracellular Cs^+ , and extracellular TEA and 4-AP) also increase the membrane resistance of the cell membrane at the holding potential to well below the series resistance, which is an important requirement for accurate membrane capacitance measurements.

If $G_{K,L}$ is indeed blocked, its gating charges should not contribute to ΔC_m , contradicting the authors' claim that transient ΔC_m can serve as a readout of $G_{K,L}$ gating.

Thank you for this valuable comment. A transient capacitance signal reflecting Na^+ channel-gating charge movement has been reported previously (Horrigan and Bookman, 1994), which could be blocked by 200 μM dibucaine, that blocks both Na^+ current (I_{Na}) and Na^+ channel-gating charge movement in squid axon (Gilly and Armstrong, 1980), but not by TTX, which blocks I_{Na} but does not immobilize Na^+ channel gating charges. Clearly TEA and 4-AP, here added to the extracellular solution to block $I_{K,L}$, together with intracellular Cs^+ , do not immobilize the related gating charges. Finding a drug that immobilizes K,L channel-gates remains an interesting task, also given the recent identification of $Kv1.8$ (*Kcna10*) channel subunits as responsible for carrying $I_{K,L}$ (Martin et al., 2023). More generally, a better knowledge of its properties is desirable given that *KCNA10* is expressed in the heart, aorta, and kidney. Very recently a missense mutation of *KCNA10* has been involved in epinephrine provoked long QT syndrome with familial history of sudden cardiac death (Huang et al., 2023). These considerations have now been added to the text (Page 15, line 431).

Moreover, the direct modulation of $I_{K,L}$ using blockers may also impact synaptic function. Given that $I_{K,L}$ is involved in the synaptic signaling at the type I hair cell–calyx synapse,

blocking these channels could alter the mechanisms the study aims to investigate, introducing additional complexities in the interpretation of the results.

Yes, direct modulation of IK,L using blockers would certainly impact the synaptic function in the case of non-quantal transmission. However, the specific goal here was to clarify if a transient exocytotic (quantal) transmission existed, as suggested by the transient change in cell membrane capacitance. The latter would have had a postsynaptic correlate in the phasic afferent response. As we demonstrated here, however, the above transient was not Ca²⁺-dependent, while it reflected the activation curve of GK,L.

That said, the study's approach is still highly informative. The comparison of ΔC_m amplitudes in wild-type and Caspr^{-/-} mice provides compelling evidence that the size of the capacitance transient is affected by changes in the synaptic cleft, with Caspr^{-/-} mice exhibiting smaller capacitance changes due to alterations in the synaptic cleft geometry.

The finding that the expression or kinetics of IK,L were not significantly altered in Caspr^{-/-} cells strengthens the argument that the differences in ΔC_m are due to changes in R_c rather than in channel function or expression.

In conclusion, while the use of blockers introduces certain limitations, the data presented in this study significantly advance our understanding of how IK,L gating regulates ΔC_m at the type I hair cell–calyx synapse. Moving forward, it would be beneficial to explore more refined techniques that can isolate the contribution of IK,L without introducing potential confounds associated with pharmacological blockers.

We thank the Reviewer for their favorable summary of our work. We do fully intend to investigate the contribution of IK,L to VHC transmission using a combination of electrophysiology and imaging techniques on transgenic mice.

Specific Comments:

I noticed some variation in the terminology used to describe hair cell types. The standard convention in scientific literature is to write "type I hair cells" and "type II hair cells" without a hyphen. I recommend maintaining consistency throughout the manuscript.

Yes, done, thank you.

There are a few typographical errors in the manuscript. For example, in the sentence: "The above findings suggest that the large transient ΔC_m recorded from type-I hair cells does not depend upon Ca²⁺ entry though voltage-gated CaV1.3 Ca²⁺ channels."

"though" should be corrected to "through."

Done, thank you.

I noticed some inconsistencies in the capitalization of the first letter in figure legends. I recommend maintaining consistency, as it is common practice to capitalize the first letter of each sentence.

Done, thank you.

Figure 2: I suggest specifying that 4-AP and TEA were added. Additionally, it would be helpful to clarify that the thick line represents the sine wave.

Done, thank you.

Regarding Figure 2B, I have a question: Why does the current appear inward at 10 mM EGTA?

We found a more representative example after reviewing the reviewer's comment and modified the figure to the version now shown. The previous unrepresentative figure could be explained as being due to a particularly small calcium current that made the potassium current appear larger than usually observed at those potentials.

Figure 3 (Panels E, F): The text states: "E, F: average differential trace obtained by subtracting ΔC_m recorded at +10 mV from that recorded at -21 mV." However, this should be corrected to: "E, F: average differential trace obtained by subtracting ΔC_m recorded at +19 mV from that recorded at -21 mV."

Thank you. We have now corrected the figure caption.

Figure 4 (Panel C): The legend states: "C: average ΔC_m recorded upon repolarization to -81 mV or -131 mV following voltage steps ranging from -101 mV to -1 mV." However, the figure itself specifies -81 mV. Could you clarify which value is correct?

Thank you. We have now corrected the figure caption.

Figure 4 (Panel D): The legend states: "D: average ΔC_m recorded upon repolarization to -131 mV following voltage steps ranging from -151 mV to -1 mV." However, earlier in the text, it is mentioned that the voltage range is -141 mV to -1 mV. Could you confirm which range is correct?

The correct value is -141 mV. This has now been corrected. Thank you.

In the section 'Does the calyx play a role in the large transient ΔC_m ?'

The sentence:

"Indeed, even the most depolarized voltage steps elicited outward (Ca^{2+}) current ≤ 100 pA in either cell types (Fig. 1A and B), meaning that R_m was at all voltages ≥ 1 G Ω , i.e. much larger than the typical R_s values." refers to Figure 1A and B, but these panels are cartoons, and the figure actually displays calcium currents. I suggest clarifying the reference to avoid confusion.

Yes sorry. It referred to Fig. 2A and B. This has now been corrected. Thank you.

Dear Dr Spaiardi,

Re: JP-RP-2025-288645R1 "Transient capacitance changes recorded from vestibular type I hair cells are produced by $G_{K,L}$ gating and do not involve neurotransmitter exocytosis" by Paolo Spaiardi, Roberta Giunta, Giorgio Rispoli, Sergio Masetto, and Stuart Leigh Johnson

Thank you for submitting your manuscript to The Journal of Physiology. It has been assessed by a Reviewing Editor and by 2 expert referees and we are pleased to tell you that it is acceptable for publication following satisfactory revision.

LANGUAGE EDITING AND SUPPORT FOR PUBLICATION: If you would like help with English language editing, or other article preparation support, Wiley Editing Services offers expert help, including English Language Editing, as well as translation, manuscript formatting, and figure formatting at www.wileyauthors.com/eoo/preparation. You can also find resources for Preparing Your Article for general guidance about writing and preparing your manuscript at www.wileyauthors.com/eoo/prepresources.

REVISION CHECKLIST:

We look forward to receiving your revised submission.

Yours sincerely,

Nathan Schoppa
Senior Editor
The Journal of Physiology

EDITOR COMMENTS

Reviewing Editor:

Comments to the Author:

The reviewers were very pleased with the revision of your paper entitled 'Transient capacitance changes recorded from vestibular type-I hair cells are produced by GK,L gating and do not involve neurotransmitter exocytosis'. There is however one minor comment from the first reviewer that remains to be addressed regarding the "cleft volume" label. Please, address this request and upload the revised version including a suitable change of label.

Senior Editor:

Comments to the Author:

The reviewers were quite pleased with the changes that you made in your revised manuscript. There remains only one minor concern from Referee 1 that should be addressed in an additional revision. That revised manuscript will not need to go out for review again; it will be checked by the reviewing editor.

REFEREE COMMENTS

Referee #1:

The authors have been responsive to comments. I have just one remaining minor criticism:

In the previous critique I suggested to "Avoid re-using labels for very different quantities ('V' is Voltage in Results and Volume in Discussion)".

The authors respond with "We replaced the cleft volume V_c with C_v throughout the text."

Forgive me, but I found this a bit amusing because C represents capacitance throughout the text - with good reason - so introducing "Cv" here is likely will likely cause as much or more confusion than "Vc". How about "Vol-CI"? It seems better to use clunky names than confusing ones, and re-using either V or C is particularly ill-advised in this specific paper (with "Cv", like "Vc", re-using both!).

Referee #2:

I appreciate the authors' thoughtful and thorough revisions. The updated manuscript significantly improves upon the original submission, both in clarity and scientific rigor.

The experimental data are robust, and the analyses are appropriate and carefully interpreted. The authors have adequately addressed the earlier concerns, and their conclusions are now well supported by the data presented.

END OF COMMENTS

Transient capacitance changes recorded from vestibular type I hair cells are produced by GK,L gating and do not involve neurotransmitter exocytosis

We appreciate the Reviewers for their valuable and insightful feedback, which has significantly improved the manuscript. We thank the Editor and Senior Editor for their comments on our manuscript. Below are our detailed responses to each of their comments.

Referee #1:

The authors have been responsive to comments. I have just one remaining minor criticism: In the previous critique I suggested to "Avoid re-using labels for very different quantities ('V' is Voltage in Results and Volume in Discussion)".

The authors respond with "We replaced the cleft volume V_c with C_v throughout the text." Forgive me, but I found this a bit amusing because C represents capacitance throughout the text - with good reason - so introducing " C_v " here is likely will likely cause as much or more confusion than " V_c ". How about " Vol_{cl} "? It seems better to use clunky names than confusing ones, and re-using either V or C is particularly ill-advised in this specific paper (with " C_v ", like " V_c ", re-using both!).

The referee is correct, and we apologize for our unclear abbreviation of the cleft volume. As suggested by the referee, we have now adopted ' Vol_{cl} ' to denote the size of the cleft volume.

Dear Dr Spaiardi,

Re: JP-RP-2025-288645R2 "Transient capacitance changes recorded from vestibular type I hair cells are produced by $G_{K,L}$ gating and do not involve neurotransmitter exocytosis" by Paolo Spaiardi, Roberta Giunta, Giorgio Rispoli, Sergio Masetto, and Stuart Leigh Johnson

We are pleased to tell you that your paper has been accepted for publication in The Journal of Physiology.

Yours sincerely,

Nathan Schoppa
Senior Editor
The Journal of Physiology

If you would like to receive our 'Research Roundup', a monthly newsletter highlighting the cutting-edge research published in The Physiological Society's family of journals (The Journal of Physiology, Experimental Physiology, Physiological Reports, The Journal of Nutritional Physiology and The Journal of Precision Medicine: Health and Disease), please click this link, fill in your name and email address and select 'Research Roundup':

<https://www.physoc.org/journals-and-media/membernews>

- You can help your research get the attention it deserves! Check out Wiley's free Promotion Guide for best-practice recommendations for promoting your work at: www.wileyauthors.com/eeo/guide. You can learn more about Wiley Editing Services which offers professional video, design, and writing services to create shareable video abstracts, infographics, conference posters, lay summaries, and research news stories for your research at: www.wileyauthors.com/eeo/promotion.

EDITOR COMMENTS

Reviewing Editor:

Comments to the Author:

It is my pleasure to inform you that your manuscript entitled „Transient capacitance changes recorded from vestibular type I hair cells are produced by GK,L gating and do not involve neurotransmitter exocytosis" has been accepted for publication in the J Physiol. We look forward to your future submission to the J Physiol.

Senior Editor:

Comments to the Author:

Your revised manuscript has addressed all of the remaining concerns and we congratulate you on its acceptance!